# Using Geostationary-Satellite-Derived Sub-Daily Fire Radiative Power Variability versus Prescribed Diurnal Cycles to Assess the Impact of African Fires on Tropospheric Ozone

Haolin Wang<sup>1,2</sup>, William Maslanka<sup>3,4,5</sup>, Paul I. Palmer<sup>2,6\*</sup>, Martin J. Wooster<sup>3,4,5</sup>, Haofan Wang<sup>1</sup>, Fei Yao<sup>2,6</sup>, Liang Feng<sup>2,6</sup>, Kai Wu<sup>7</sup>, Xiao Lu<sup>1\*</sup>, Shaojia Fan<sup>1\*</sup>

<sup>1</sup>School of Atmospheric Sciences, Sun Yat-sen University, and Southern Marine Science and Engineering Guangdong Laboratory (Zhuhai), Zhuhai, China

Correspondence to: Paul I. Palmer (pip@ed.ac.uk), Xiao Lu (luxiao25@mail.sysu.edu.cn), and Shaojia Fan 15 (eesfsj@mail.sysu.edu.cn)

Abstract. Assessing the impact of biomass burning (BB) emissions on tropospheric ozone is critical for understanding air pollution and climate interactions. BB emission inventories like Global Fire Emissions Database and Global Fire Assimilation System, typically based on sun-synchronous satellite observations, report emissions on daily, weekly or longer timescales with empirically derived factors generally used to overlay diurnal variations. To explore the sensitivity of tropospheric ozone to diurnal variability, we incorporated day-specific hourly BB variations inferred from geostationary satellite data into the GEOS-Chem atmospheric chemistry transport model. The simulations were compared with those using established inventories and evaluated against in situ and satellite observations. Simulations with real hourly-resolved emissions produce comparable surface ozone biases (-1.54 to +9.09 ppbv vs. -1.58 to +9.13 ppbv) and marginally higher correlations with TROPOMI nitrogen dioxide (r = 0.80-0.89) and OMI ozone (r = 0.80-0.94). Although the statistical improvements are limited, the geostationary-driven approach reveals pronounced regional ozone differences and mechanistic insights into the role of diurnal fire variability. Data-driven diurnal BB variations across Africa cause significant surface ozone changes (-8.57 to +21.88 ppbv) and alter tropospheric ozone columns by -0.41 to 1.09 DU, particularly in regions with intense fire activity like Angola and Zambia. These changes propagate globally, shifting regional OH concentrations by -4.4% to +51.7%. These findings emphasize the critical role of accurately describing diurnal BB variations in atmospheric models to better quantify its impacts on atmospheric composition, providing insights for Earth system model development and the use of geostationary-derived BB emissions datasets.

<sup>&</sup>lt;sup>2</sup>School of GeoSciences, University of Edinburgh, Edinburgh, UK

<sup>&</sup>lt;sup>3</sup>Department of Geography, King's College London, London, UK

<sup>&</sup>lt;sup>4</sup>National Centre for Earth Observation, King's College London, London, UK

<sup>&</sup>lt;sup>5</sup>Leverhulme Centre for Wildfires, Environment and Society, King's College London, London, UK

<sup>&</sup>lt;sup>6</sup>National Centre for Earth Observation, University of Edinburgh, Edinburgh, UK

<sup>&</sup>lt;sup>7</sup>Department of Civil and Environmental Engineering, University of California, Irvine, CA, USA

## 1 Introduction

Biomass burning (BB) – the large-scale combustion of live and dead organic plant matter – is a critical process in the Earth system, influencing ecosystem functioning by recycling nutrients and enabling serotiny. However, its incomplete combustion releases significant quantities of gaseous and particulate pollutants alongside CO<sub>2</sub>, including greenhouse gases like methane and nitrous oxide, and reactive species such as carbon monoxide (CO), nitrogen oxides ((NO<sub>x</sub> = NO + NO<sub>2</sub>)), and volatile organic compounds (VOCs) (Koppmann et al., 2005; Reid et al., 2005; Andreae, 2019). These emissions drive regional and global tropospheric chemistry, contributing to the production of tropospheric ozone, a key air pollutant and greenhouse gas with implications for air quality, human health, and climate (Parrington et al., 2013; Bourgeois et al., 2021; Xu et al., 2023; Marvin et al., 2024). Globally, wildfires are estimated to produce approximately 170 Tg of ozone annually, accounting for 3.5% of total tropospheric ozone production (Jaffe and Wigder, 2012).

Due to its unique biophysical and climatic conditions, the African continent has long been a global hotspot of BB activity (Archibald et al., 2012), recognised in the scientific literature since the 19<sup>th</sup> Century (von Danckelman, 2009). African BB emissions, primarily in tropical grasslands, savannas, and shrublands, account for ~70% of the global burned area and ~50% of fuel consumption, driven by abundant vegetation, stable dry seasons, and frequent natural and anthropogenic ignition sources (Bowman et al., 2020; Jiang et al., 2020). Southern Africa has been the focus of extensive field campaigns investigating biomass burning emissions, with prominent dry-season initiatives including the Southern African Fire-Atmosphere Research Initiative (SAFARI-92; Trollope et al. (1996)) and the South African Regional Science Initiative (SAFARI 2000; Swap et al. (2003)). By integrating in situ measurements with atmospheric chemistry transport models, these studies have revealed key aspects of biomass burning, including the magnitude and distribution of regional emissions, the atmospheric transport processes governing their dispersion, and their combined effects on air quality and climate. Previous research has identified Africa as the most significant contributor to tropospheric ozone associated with biomass burning, accounting for approximately 35% of the global annual pyrolytic ozone enhancement (Marufu et al., 2000). Data from the Ozone Monitoring Instrument (OMI) and the Microwave Limb Sounder (MLS) aboard the NASA Aura satellite, coupled with outputs from an atmospheric chemistry transport model, highlighted a persistent enhancement in tropospheric ozone of 5-8 ppbv, above typical values of 35-55 ppby, over sub-Saharan Africa during fire seasons (Ziemke et al., 2009). Similarly, the column ozone amount increased by approximately 10-20%. In addition, Jourdain et al. (2007) employed the GEOS-Chem atmospheric chemistry transport model to link BB over West and Central Africa to ozone-enriched layers observed by the NASA Tropospheric Emission Spectrometer aboard the Aura satellite, contributing to increases in the lower tropospheric ozone background by 20-50%. These findings underscore Africa's critical role in the global tropospheric ozone budget and the need for accurate BB emission estimates to inform climate and air quality strategies.

Previous studies have characterized the influence of the magnitude and seasonal variation of African BB emissions on tropospheric ozone. However, critical gaps remain in understanding regional-scale ozone production from African fires,

especially about including data-driven changes in diurnal emission patterns in atmospheric transport models. Most atmospheric modelling studies simplify biomass burning emissions by assuming temporally uniform release rates (Jourdain et al., 2007; Ziemke et al., 2009), an approximation that may obscure important short-term chemical and transport dynamics. Fire behaviour—encompassing speed, intensity, and spatial distribution—is governed by a complex synergy of meteorological factors that vary diurnally (e.g., wind, temperature, and atmospheric stability), fuel characteristics (e.g., moisture content, biomass density, and vegetation type), and ignition sources (Archibald et al., 2013; Bowman et al., 2020). Furthermore, the spatiotemporal distribution of fires is intrinsically linked to broader environmental and anthropogenic drivers, including climatic variability, human land-use decisions, and regional fire governance strategies (Lavorel et al., 2007; Kelley et al., 2019). The interplay of these factors not only determines fire frequency and severity but also imposes a pronounced diurnal signature on fire activity. For instance, daytime heating and lower humidity enhance fire intensity and emissions, while nighttime cooling suppresses combustion, creating dynamic emission profiles rarely captured in conventional models or reliant on fixed diurnal profiles (Heald et al., 2003; Zhu et al., 2018). More recently, Tang et al. (2022) emphasized that fire diurnal cycles exert strong influences on regional air quality during major field campaigns in the United States, underscoring the need for improved representation of sub-daily variability in chemical transport models (Mu et al., 2011; Li et al., 2019). This highlights a critical research need: understanding how the observed diurnal variability in African biomass burning emissions affects the tropospheric ozone budget both within Africa and in downwind regions.

BB emission inventories, typically built from satellite-derived burned area or active fire counts combined with fuel load estimates and emission factors (Seiler and Crutzen, 1980; Andreae and Merlet, 2001), have improved significantly through further advancements in the spatial and temporal resolution of satellite observations, numerical modelling and data assimilation, as well as in field campaigns (Giglio et al., 2018; Ichoku, 2020; Xu and Wooster, 2023). However, uncertainties persist in quantifying emissions (e.g., magnitude, patterns, vertical injection height), potentially introducing errors in regional and global air pollutant distributions (Reddington et al., 2016; Petrenko et al., 2017). Most widely used global fire emission inventories, such as Global Fire Emission Database (GFED), Global Fire Assimilation System (GFAS), Quick Fire Emissions Dataset (QFED), and Fire Inventory from NCAR (FINN), provide emissions at monthly or daily resolution. GFED is based on the MODIS burned area product available at monthly intervals, while GFAS, OFED, and FINN use active fire detections or FRP to provide daily emissions that can be further disaggregated to sub-daily scales using empirical or observation-based factors (WRAP, 2005; Akagi et al., 2011; Wooster et al., 2021). This approach introduces uncertainties in assessing short-term emission variability and its impact on atmospheric composition. This limitation underscores the need for higher-resolution data to better characterize short-term emission variability and its atmospheric effects. Previous efforts have sought to characterize fire diurnal cycles directly from satellite observations, for example by combining geostationary and polar-orbiting data for Africa (Freeborn et al., 2009) and by developing new MODIS-based diurnal parameterizations to improve FRP-derived fire energy estimates (Andela et al., 2015). Addressing these uncertainties is particularly critical in regions with significant BB activity, where changing fire patterns further complicate emission estimates. This highlights the critical importance of advancing African BB research is vital for developing climate-adaptive strategies that address both local environmental management and global carbon budgeting challenges (Walker et al., 2019; IPCC, 2023).

In this study we aim to develop a better understanding of how real diurnal variations in African BB affect tropospheric ozone production and atmospheric oxidation capacity. We compare the results to those obtained when a fixed diurnal cycle is assumed. We use an inventory derived using the 'Fire Radiative Energy eMissions (FREM)' approach to prescribe the emissions (Nguyen and Wooster, 2020), whereby Fire Radiative Power (FRP) timeseries derived from geostationary data collected by the Meteosat Spinning Enhanced Visible Infra-Red Imager (SEVIRI; Wooster et al., 2015) is combined with carbon monoxide column data from the TROPOspheric Monitoring Instrument (TROPOMI) aboard the ESA Sentinel-5 Precursor (Sentinel-5P) satellite to infer rates of biomass burning change across each day analysed (Wooster et al., 2015). These FREM-derived 'top-down' BB emissions for Africa are described in (Nguyen et al., 2023), and here we have expanded this inventory using additional emissions factors to include the full suite of BB emissions used by the GEOS-Chem model, allowing us to evaluate the impact of observed diurnal emission variations on describing the observed distribution of tropospheric ozone. Section 2 describes the data and methods in more detail; outlining the GEOS-Chem Atmospheric Chemistry Transport Model, as well as the satellite observations used in this study. Section 3 outlines the results and analysis, and in Section 4 we provide a summary and conclude the study.

## 2 Data and Methods

Here we describe the GEOS-Chem atmospheric chemistry transport model and the satellite data to explore the impact of diurnal variations in BB emissions in Africa, inferred from geostationary satellite data, on the regional and global chemical composition of the atmosphere. For this study, we focus on four months in 2019 that exemplify seasonal changes over Africa: January, April, July, October.

## 2.1 The GEOS-Chem Atmospheric Chemistry Transport Model

We employ the global three-dimensional chemical transport model GEOS-Chem version 13.3.1 (available at https://github.com/geoschem/GCClassic/tree/13.3.1, last accessed 8 February 2025) to interpret the impact of diurnal variations in BB emissions in Africa on the chemical composition of the atmosphere. The model is driven by Goddard Earth Observing System, version 5 (GEOS-5) Modern-Era Retrospective analysis for Research and Application version 2 (MERRA-2) assimilated meteorological fields from the Global Modeling and Assimilation Office (GMAO) at NASA Goddard Space Flight Center. We perform nested-grid simulations over whole Africa (18.75°W -52.0°E, 36.0°S-18.0°N), also driven by the MERRA-2 assimilated meteorological fields at a spatial resolution of 0.5°(latitude) × 0.625°(longitude). We spin up the GEOS-Chem model by one month for each of the four typical months of simulation. Time-dependent lateral boundary conditions are

archived from the global simulation run at a coarser resolution of  $2^{\circ}$  latitude  $\times 2.5^{\circ}$  longitude. For both models, we use 47 hybrid-sigma levels extending from the surface to 0.01 hPa.

The GEOS-Chem model incorporates an extensive chemical mechanism for stratospheric and tropospheric ozone–NO<sub>x</sub>–VOCs–aerosol–halogen interactions, as outlined by Eastham et al. (2014). The model utilizes chemical kinetics data from the Jet Propulsion Laboratory (JPL) and the International Union of Pure and Applied Chemistry (IUPAC), as described by (Burkholder et al., 2015) and IUPAC (2013), respectively. Photolysis rates are computed using the Fast-JX scheme (Bian and Prather, 2002). Tracer advection is achieved through the utilization of the TPCORE advection algorithm (Lin and Rood, 1996), while the boundary layer mixing process is characterized by the non-local scheme (Lin and McElroy, 2010). Dry deposition of gases and aerosols is calculated using the resistance-in-series algorithm (Wesely, 1989; Zhang et al., 2001). Wet deposition of water-soluble aerosols and gases is described by the methodologies presented by Liu et al. (2001) and Amos et al. (2012), respectively.

Emissions in GEOS-Chem are included using the Harvard-NASA Emission Component (HEMCO) (Keller et al., 2014). We use the latest version of the Community Emissions Data System (CEDSv2) anthropogenic emissions inventory (O'Rourke et al., 2021). Online calculation of natural emissions of biogenic VOCs (Guenther et al., 2012), lightning  $NO_x$  (Murray et al., 2012), and soil  $NO_x$  (Hudman et al., 2012) are used in both global and regional simulations. We refer the reader to Wang et al. (2022) for more details on the model configurations. BB emissions are also incorporated in our simulations. Detailed descriptions of the BB inventories, including the fire radiative energy emission inventory, can be found in Sections 2.2 and 2.4.

## 2.2 Fire Radiative Energy Emission Inventory

- The Fire Radiative Energy eMission (FREM) inventory is a "direct" and "top-down" style of BB emission inventory, in that it is solely based on satellite remote sensing observations, without the need for modelling or additional assumptions. This is in direct contrast to non-direct, more "bottom-up" inventories, such as the GFED (Randerson et al., 2017), whose emissions are estimated by combining burnt-area extent observations with combustion completeness assumptions and pre-fire fuel load estimations (Reid et al., 2009).
- The first iteration of the FREM approach (FREMv1) focused on deriving Total Particulate Matter (TPM) emissions from FRP observations (Mota and Wooster, 2018; Nguyen and Wooster, 2020; Nguyen et al., 2023), using aerosol optical depth (AOD) estimations to calculate plume-integrated Total Column PM<sub>2.5</sub> amounts. This was done via a training dataset of fire-matchups of manually identified and digitized plume extents from fires and their resulting smoke plumes, subject to cloud-free conditions and sufficient near-surface winds to aid in the identification of smoke plumes in the AOD observations. FREMv1 was enhanced by (Mota and Wooster, 2018; Nguyen and Wooster, 2020; Nguyen et al., 2023) to improve the performance, via improved AOD datasets. The current version of FREM (FREMv2, (Mota and Wooster, 2018; Nguyen and Wooster, 2020; Nguyen et

al., 2023) replaced the AOD dataset with Total Column Carbon Monoxide (TCCO) observations to directly estimate CO fluxes, rather than TPM. However, both FREMv1 and FREMv2 use geostationary FRP data owing to its high temporal resolution.

Both iterations of the FREM approach follow the same methodology; by grouping the fire-matchups total Fire Radiative Energy (FRE) and plume-integrated totals of emissions of TPM and/or CO by their respective biomes, a series of biomespecific emission coefficients (Ce) were generated, directly relating FRE to the emitted TPM and/or CO. These derived biomespecific emission coefficients can then be used with the entire SEVERI-derived FRP timeseries to calculate the emissions of TPM and/or CO at the native temporal resolution of 15 minutes across the entire timeseries. Additionally, biome-specific emission coefficients of other species can be estimated using emission factor (EF) ratios, via Equation (1), allowing for the generation of other emission timeseries:

$$Ce_y^{biome} = \left[ \frac{(EF_y^{biome})}{(EF_x^{biome})} \right] Ce_x^{biome}, \tag{1}$$

where  $Ce_y^{biome}$  and  $EF_y^{biome}$  are the biome-specific emission coefficient (g MJ<sup>-1</sup>) and emission factor (g kg<sup>-1</sup>) of the target species y, while  $Ce_x^{biome}$  and  $EF_x^{biome}$  are the biome-specific emission coefficient (g MJ<sup>-1</sup>) and emission factor (g kg<sup>-1</sup>) of the reference species x (typically TPM or CO). Comparisons to fire emissions data from the GFEDv4 between 2004 and 2020 across Africa showed regional CO emissions are remarkably close, despite the two inventories being independent and utilizing different satellite datasets and methodologies (Nguyen et al., 2023).

For this study, we used the FREM biome-specific emission coefficient for CO (Nguyen et al., 2023), alongside the geostationary Meteosat SEVERI FRP-GRID data product (Wooster et al., 2015) at its native spatio-temporal resolution (15 minute, 0.1° x 0.1°) across the African continent (40°N - 37°S, 20°W - 52.9°E) between 1st January and 31st December 2019, generating an African wide timeseries of gridded CO emissions. To assess the impact of diurnal variations in fire emissions, the FREM CO emissions were then aggregated into three different temporal resolutions: hourly (FREM\_hourly), tri-hourly (FREM\_3hourly), and daily (FREM\_daily), each giving gridded summations of CO at these three resolutions.

# 2.3 In situ Observations

We use surface, sonde, and aircraft in situ measurements of ozone over Africa to help evaluate our GEOS-Chem model simulation. Surface ozone observations are obtained from the South African Air Quality Information System (SAAQIS, <a href="https://saaqis.environment.gov.za/">https://saaqis.environment.gov.za/</a>) for 2019. Following the data quality control procedures established by Wang et al. (2023) and Wang et al. (2024), we apply rigorous filtering to exclude unreliable measurements. There is a total of 84 sites used for evaluation during the study period. For upper-tropospheric and lower-stratospheric ozone validation, we utilize ozonesonde data from the Southern Hemisphere ADditional OZonesondes network (SHADOZ, <a href="https://doi.org/10.57721/SHADOZ-V06">https://doi.org/10.57721/SHADOZ-V06</a>). SHADOZ, operational since 1998, comprises 14 tropical and

subtropical stations providing high-vertical-resolution ozone profiles widely used for satellite validation and model evaluation, particularly in the launch up to 35 km altitude range (Thompson et al., 2017; Witte et al., 2017; Sterling et al., 2018; Witte et al., 2018). The profiles are collected from electrochemical concentration cell-type sensors launched with a standard radiosonde and are the reprocessed/homogenized V06 data described by Witte et al. (2017). In this study, we focus on two African stations: Ascension Island and Nairobi. Recent research confirmed the long-term stability of SHADOZ measurements, demonstrating consistent agreement with satellite total column ozone observations and independent stratospheric ozone records over the past 18 years (Stauffer et al., 2022). This established reliability supports the use of SHADOZ data as a robust benchmark for model evaluation (Thompson et al., 2021; Wang et al., 2022).

We also use aircraft measurements from In-service Aircraft for a Global Observing System (IAGOS, <a href="https://www.iagos.org/">https://www.iagos.org/</a>). IAGOS is a European Research Infrastructure initiated in August 1994 for global observations of atmospheric composition from instruments on board commercial aircraft of internationally operating airline (Thouret et al., 1998; Nédélec et al., 2015). Ozone is measured by ultraviolet absorption monitor at 253.7 nm with a time resolution of 4s, a precision of ±2%, and an accuracy of about ±2 ppbv. Previous studies have established the reliability of IAGOS ozone data through extensive validation, showing close agreement with ozonesonde measurements (Staufer et al., 2013, 2014) and demonstrating its capability to accurately represent lower tropospheric ozone variability (Petetin et al., 2018; Cooper et al., 2020). These findings underscore the suitability of IAGOS observations for assessing regional-scale tropospheric ozone trends, particularly in the free troposphere where such high-resolution aircraft measurements provide critical constraints. Here we focus on the Africa region in 2019 to assess the performance capability of the model. Figure 2 shows a total of 131 available ozone profiles are obtained across Africa during the four study months (January, April, July, and October) of 2019. We use a consistent methodology to sample the GEOS-Chem simulations, driven by the different inventories, at the time and location of each observation, including along flight tracks and ozonesonde trajectories.

## 2.4 Satellite Observations

We also use satellite column observations to help evaluate our GEOS-Chem model because they provide self-consistent, continental-scale information that complements the in situ data. We use total CO columns and tropospheric NO<sub>2</sub> columns during 2019 retrieved from TROPOMI, which is a nadir-viewing 108-degree field-of-view, push-broom grating hyperspectral spectrometer (Veefkind et al., 2012) onboard the Sentinel-5P satellite with an ascending local equator crossing time of 13:30. TROPOMI collects spectrally resolved data at ultraviolet-visible (UV-VIS, 270nm to 495nm), near-infrared (NIR, 675nm to 775nm), and shortwave infrared (SWIR, 2305nm to 2385nm) wavelengths. It achieves near-daily global coverage by having an across-track swath width of 2600 km and has a ground footprint spatial resolution of 5.5 × 7.0 km² for CO and NO<sub>2</sub> at nadir. We refer the reader to dedicated studies that describe the retrieval of CO (Vidot et al., 2012; Landgraf et al., 2016) and NO<sub>2</sub> (van Geffen et al., 2020). We use TROPOMI tropospheric ozone data retrieved by the convective-cloud-differential (CCD) algorithm (Valks et al., 2003; Valks et al., 2014). This data product is gridded daily at a resolution of 0.5° latitude by 1°

longitude resolution between 20°S and 20°N, and represents a 3-day moving mean tropospheric ozone column between the surface and 270 hPa in cloud-free conditions (Hubert et al., 2021). In our study, we only used TROPOMI satellite retrievals associated with quality assurance flags > 0.5 for CO, >0.75 for NO<sub>2</sub>, and >0.5 for ozone, which corresponds to good-quality retrievals over nearly cloud-free scenes, as recommended by the relevant technical notes (available at https://sentinels.copernicus.eu/web/sentinel/technical-guides/sentinel-5p/products-algorithms; last accessed 20 July 2023).

We also include monthly OMI/MLS tropospheric ozone columns with resolutions of 1° × 1.25° for 2019 and latitude range 60°S–60°N. OMI/MLS tropospheric ozone is determined daily by subtracting the co-located MLS stratospheric ozone column from the OMI total ozone column (Ziemke et al., 2006). Monthly mean fields are then determined by averaging all available daily data within each month. More details for the OMI/MLS tropospheric ozone columns data are described in Ziemke et al. (2019).

To compare the model against TROPOMI, GEOS-Chem CTM is sampled at the time and location of each TROPOMI measurement to generate vertical profiles x to which we then apply a scene-dependent instrument averaging kernel (A) that describe the instrument vertical sensitivity:

$$\widehat{\mathbf{x}} = A\mathbf{x} + (1 - A)\mathbf{x}_{a},\tag{2}$$

where  $x_a$  denotes a priori vertical profile from TROPOMI; lower- and upper-case variables in bold denote vector and matrix quantities, respectively. We convert the simulated profile to partial column, following Zhang et al. (2010).

# 2.5 Numerical Experiments

To assess the impact of geostationary satellite data-driven, day-specific diurnal varying BB emissions on tropospheric ozone within GEOS-Chem model (Table 1), we use the FREM African BB emissions inventory for CO of Nguyen et al. (2023), extend it to multiple emitted species using biome-specific emissions factors presented therein, and aggregate it to three different temporal resolutions (hourly, three-hourly, and daily). We compare this inventory to the two most commonly used BB inventories – the Global Fire Emissions Database Version 4.1 (GFED4.1s) and the GFAS (Kaiser et al., 2012). In GFED4.1s, emissions of trace gases and aerosols are derived using a bottom-up approach that combines MODIS burned area, fuel load, combustion completeness, and biome-specific emission factors defined per unit of dry matter burned (Randerson et al., 2017; van der Werf et al., 2017). GFAS follows a similar principle, converting MODIS FRP into dry matter combustion using land-cover-specific FRP-to-dry-matter-consumed factors, and then deriving emissions using biome-dependent emission factors (Kaiser et al., 2012). By contrast, the FREM inventory provides only CO emissions derived from geostationary FRP and Sentinel-5P CO observations (Nguyen et al., 2023); other species (e.g., NO<sub>x</sub>, VOCs, aerosols) are obtained by scaling CO using biome-specific CO-to-species emission factor ratios (Andreae and Merlet, 2001; Akagi et al., 2011; Andreae, 2019).

The GFED4.1s dataset provides monthly data on burned area, fire carbon, and dry matter (DM) emissions, along with the contributions from different fire types, enabling the calculation of trace gas and aerosol emissions via emission factors. GFED4.1s inventory includes fractional contributions from various fire types, with daily or 3-hourly emissions which are scalar fields to increase the temporal resolution of monthly emissions based on active fire distributions and diurnal cycles informed by climatological data. Importantly, GFED4.1s integrates small fire inputs to improve the precision and comprehensiveness of emission estimates (Randerson et al., 2017; van der Werf et al., 2017). The GFAS inventory estimates dry matter burning through FRP observations from the MODIS instruments onboard the Terra and Aqua satellites, offering daily spatially gridded BB emissions for a wide range of chemical species, greenhouse gases, and aerosols at a spatial resolution of  $0.1^{\circ} \times 0.1^{\circ}$  (Kaiser et al., 2012).

We prepare two baseline simulations and a total of six sensitivity simulations to investigate the impact of diurnal BB emissions variations in Africa on the regional and global chemical composition of the atmosphere, described in Table 1. In all simulations we use the same anthropogenic and non-BB natural emissions. To reiterate, all GEOS-Chem simulations are driven by hourly BB emissions, but this hourly emissions information in the case of GFED and GFAS is derived from coarser temporal resolution emissions data interpolated using their climatological diurnal scaling factors. Our FREM-derived emissions on the other hand, is based on observed emissions variability derived from real changes in FRP recorded every 15-minutes in the Meteosat SEVIRI-derived data.

Our self-consistent global (GBASE) and nested (NBASE) baseline simulations use the GFED4 BB inventory. For both baseline simulations, we use daily fire emission fractions and diurnal cycles that allocate daily emission estimates over eight three-hour windows during a day (GFED4\_3hourly). This implies that the emission remains constant over those three-hour intervals, resulting in only eight daily updates. Our GBASE simulation provides the three hourly lateral boundary conditions necessary to run our NBASE simulation over the African domain (defined above). We also include another nested simulation based on daily BB emissions estimates from the GFAS inventory (GFAS\_daily), which uses the same diurnal scaling factors, derived from the Western Regional Air Partnership based on data collected over the United States (WRAP, 2005), for every day and every grid point affected by BB. Using this approach, peak emissions occur during daytime hours, between 10:00 and 19:00 local time, with significantly lower values during the remainder of the day and night. This uniform temporal profile of wildfire emissions has been adopted by other major BB emission inventories, including the Quick-Fire Emissions Dataset (QFED) and the historical biomass burning emissions for CMIP6 (BB4CMIP).

To assess the impacts of time-resolution of BB emission inventories on atmospheric composition, we used the hourly (FREM\_hourly), three-hourly (FREM\_3hourly), and daily (FREM\_daily) versions of the FREM inventory. Fire plumes can inject emissions above the planetary boundary layer (PBL), and for FREM we use smoke plume injection height assumptions consistent with those in GFAS. GFAS employs two models, the Plume Rise Model (PRM) and the Integrated Monitoring and Modelling System for Wildland Fires (IS4FIRES), to estimate injection heights. These calculations integrate satellite-derived

Fire Radiative Power (FRP) data with ECMWF forecasts of critical atmospheric variables (Rémy et al., 2017). For GFED4, we follow the approach of Fischer et al. (2014) and Travis et al. (2016), allocating 65% of biomass burning emissions to the PBL and the remaining 35% to the free troposphere. For FREM\_daily, we use the fixed diurnal variation factor used by GFAS\_daily, as described above. The comparison between NBASE and FREM\_3hourly and between GFAS\_daily and FREM\_daily helps us to assess the impact of different BB emissions inventories, with different emissions totals but the same temporal resolution, on model output. The comparison of FREM\_hourly with FREM\_daily helps us assess the impact of a day-specific diurnal cycle in BB emissions on atmospheric composition. In a final set of calculations, we include a coarser version of FREM\_hourly and FREM\_daily in a simulation with a global spatial resolution of  $2^{\circ} \times 2.5^{\circ}$  – these are denoted as GFREM\_hourly and GFREM\_daily, where the leading G in the acronym denotes global. The difference between these two calculations allows us to further investigate the impact of the diurnal cycle of BB emissions in Africa on the global chemical composition of the atmosphere.

# 290 **2.6 Ozone budget diagnosis**

To investigate the impact of the diurnal varying BB emissions on ozone, we employ budget diagnostics to assess the roles of four key processes influencing ozone concentrations: chemistry, transport (including horizontal and vertical advection), mixing, and convection. We do not include a description of the effects of dry deposition here due to the non-local Planetary Boundary Layer (PBL) mixing scheme used in the model, which incorporates the dry deposition process within the mixing mechanism. GEOS-Chem v12.1.0 and later versions offer budget diagnostics, defined as the mass tendencies (kg s<sup>-1</sup>) for each grid cell of each species in the column (total, tropopause, and PBL). These diagnostics reflect the changes in vertically integrated column ozone mass before and after accounting for chemistry, transport, mixing, and convection components, providing a more detailed perspective on how these processes interact for ozone changes in the diurnal variation of BB emissions.

## **300 3 Results**

#### 3.1 Importance of the Diurnal Cycle of Biomass Burning Emissions on Atmospheric Composition

CO emissions often are used as indicators to evaluate and compare differences among various BB inventories (Hua et al., 2024). However, here we are most interested in investigating the impact of BB on tropospheric ozone, so we focus on  $NO_x$  emissions as it is a key precursor for ozone formation. Previous studies have primarily compared the spatial distribution and total emissions of BB (Reddington et al., 2016; Carter et al., 2020; Nguyen et al., 2023; Jin et al., 2024), but here we extend this to include analysis of the diurnal cycle differences among the three BB emission inventories studied (FREM, GFED and GFAS). Figure 1 illustrates for 2019 the diurnal variations for four representative months for each inventory (top row), the total monthly  $NO_x$  emissions for each month (middle row), and finally the spatial distribution of  $NO_x$  emissions (bottom rows).

In all datasets, African BB  $NO_x$  emissions peak in January and July, corresponding to fire seasons in Northern Hemisphere Africa and Southern Hemisphere Africa respectively and focused on in Figure 1. For comparison, we also present emissions for April and October, which represent months of relatively low fire activity within the fire season. All three BB inventories (FREM, GFED4.1, GFAS) show consistent seasonal variations in  $NO_x$  emissions, but there are large differences in their total emissions – mainly in January and July for northern and southern hemisphere Africa, respectively. The highest total  $NO_x$  emission is reported by GFED4.1 (peak of 1318.3 Gg in July), ~275% higher than GFAS and ~159% higher than FREM. This is consistent with the recent study of Wang et al. (2025)who also suggested that GFED4 may overestimate  $NO_x$  emissions from African BB.

Figure 1 diurnal cycles of BB NO<sub>x</sub> confirm that emissions typically are far higher during the day than at night. This behaviour is described by the geostationary-derived FREM inventory as expected due to its hourly temporal resolution based on observations but seems also reasonably captured by the diurnal scaling factors applied in both the GFED4 and GFAS inventories. However, our analysis reveals significant discrepancies between the 'true' FREM hourly inventory, and the daily totals from the FREM inventory adjusted to be hourly using the idealized GFAS diurnal cycle empirical factors. The FREM daily resolved emission inventory exhibits higher daytime peaks than the hourly inventory in January, while the opposite is observed in July, with peak NO<sub>x</sub> emission differences reaching up to 0.43 Gg hour<sup>-1</sup>. During July, the peak emission timing differs between the FREM daily and hourly inventories, occurring at 13:00 and 12:00 local time respectively. During seasons of lower wildfire activity, such as April and October, the differences between the daily and hourly FREM inventories are much smaller. In terms of all three inventories, during the fire season, GFAS consistently shows the lowest emissions, and GFED4 the highest. GFED4.1 exhibits higher daytime hourly emissions rates compared to FREM or GFAS, with NO<sub>x</sub> emission differences reaching up to 3.58 Gg hour<sup>-1</sup>.

A similar comparison for CO emissions is presented in Figure S1. CO is the principal species used for intercomparison across multiple biomass burning emission inventories (Pan et al., 2020; Jin et al., 2024) and serves as a key tracer of combustion and long-range transport (Duncan et al., 2003; Edwards et al., 2006). All three inventories capture the pronounced north—south seasonal contrast of African fires, with peak activity in January and July and relatively weak emissions in April and October. The total emitted CO differs noticeably among inventories, with GFED4 generally reporting higher regional totals than the others, while the relative differences vary somewhat between months and regions. The diurnal variation of CO emissions is broadly consistent with that of  $NO_x$ , showing a clear daytime maximum around local noon and minima at night, but the amplitude and peak timing vary among inventories, reflecting differences in the representation of fire activity and emission factors.

Figure 1 also shows the monthly distribution of BB emissions of  $NO_x$  during January, April, July, and October 2019. The different inventories report substantial differences in the magnitude and spatial distributions of these emissions, which may

have profound impacts on the subsequent atmospheric chemistry. The differences in the diurnal distribution of these emissions discussed above may also exacerbate this situation. To investigate these impacts, we analysed the outputs of our GEOS-Chem atmospheric chemistry transport model runs.

# 3.2 Impact of BB Emissions on GEOS-Chem Simulations and Model Evaluation

In Figure 2 we evaluate our model simulations using multiple observational datasets. Figure 3 and Table 2 shows the mean bias between the model simulations and all in situ observations for our four study months. Taylor statistics, which incorporate three key metrics, centred root-mean-square error, correlation coefficients, and normalized standard deviation, are used to compare model outputs with satellite retrievals (Figures 4 and S2). Table S1 synthesizes the validation results by presenting correlation coefficients and mean biases between model simulations and all satellite products. Spatial distributions of these atmospheric components are further illustrated in Figures S3—S6.

# 350 Model Evaluation using In situ Observations

Using NBASE as our reference model simulation, we find that the GEOS-Chem model captures the essential features of tropospheric ozone vertical distributions across tropical regions, demonstrating good consistency with IAGOS aircraft and ozonesonde measurements in absolute ozone concentrations (Figure 2). Model-observation comparisons reveal a persistent -2.38 to 5.21 ppbv bias in the free troposphere relative to ozonesonde data, consistent with IAGOS data (-1.25 to 7.74 ppbv), as shown in Figure 3. Specifically, the model generally overestimates ozone concentrations (1.77-7.74 ppbv) compared to IAGOS aircraft observations, except for northern Africa in July where it underestimates ( $-1.25 \pm 6.07$  ppbv). At the Nairobi ozonesonde station, negative model biases occur in April and October, while ozone is overestimated throughout the 1000-200 hPa layer during other months – a pattern that contrasts with the Ascension site where positive biases persist consistently across all four observed months. Surface ozone comparisons at 84 South African monitoring sites during our four study months in 2019 show the model reproduces observed temporal variations with mean biases ranging from -0.93 to 8.77 ppbv and correlation coefficients between 0.50 and 0.65. The model exhibited the highest positive bias in January (8.77  $\pm$  12.94 ppbv), followed by October (7.96 ± 11.81 ppbv) and April (5.82 ± 11.63 ppbv), while July showed near-neutral performance with a slight negative bias ( $-0.93 \pm 10.66$  ppbv). The tendency of the model to overestimate peak concentrations likely stems from two factors – recent updates to chemical mechanisms incorporating heterogeneous NO<sub>v</sub> chemistry in aerosols and clouds (Holmes et al., 2019) and revised oceanic ozone deposition schemes (Pound et al., 2020), combined with potential overestimations of anthropogenic emissions in developing tropical regions (Wang et al., 2022).

Figure 3 and Table 2 show the mean statistical comparison, including biases with their standard deviations of our five nested model runs (Table 1). Our reference model, NBASE, has the largest positive tropospheric mean biases (1.77–7.74 ppbv for IAGOS and 2.16–5.21 ppbv for ozonesonde) and the largest surface biases (5.82–8.77 ppbv) of all the model runs for all our

study months, particularly evident during January and April over northern Africa, throughout all months over southern Africa, and in January/July at the Nairobi ozonesonde station, which is consistent with results reported by Wang et al. (2025) for this region. GFAS also performs well in reproducing the observational data, particularly the aircraft data in the southern hemisphere below 400 hPa and the Nairobi ozonesonde data over the range 750-400 hPa. FREM shows comparable performance with GFAS in the tropospheric ozone simulation, with no statistically significant differences. The model ozone bias ranges show 375 close agreement between inventories: -2.40 to 7.88 ppbv for GFAS compared with -2.47 to 7.73 ppbv for FREMs. This consistency extends to surface ozone comparisons, where both inventories again demonstrate remarkably similar bias ranges (-1.70 to 8.89 ppby for GFAS versus -1.58 to 9.13 ppby for FREM). Our analysis of different temporal resolutions in FREM reveals that the hourly (FREM hourly) and 3-hourly (FREM 3hourly) version generally produces smaller surface biases (-1.54 to 9.09 ppby) compared to daily (-1.58 to 9.13 ppby) versions. The exceptions are the comparison with the ozonesondes - that get progressively worse with higher temporally resolved inventories, especially in January at the Ascension site and in 380 April at the Nairobi site. However, these differences are not statistically significant as shown in Table 2. This limited improvement from finer temporal resolutions is likely due to the minimal contribution of local fire sources at several in situ sites, where long-range atmospheric transport acts as a low-pass smoothing filter due to diffusive mixing (Mu et al., 2011), thus constraining the benefits of replacing daily, weekly or longer timescales emissions with hourly fire emissions data.

# Model Evaluation using Independent Satellite Observations

The model accurately reproduces the spatial distribution of total column CO observed by TROPOMI in all four months (Figure S3). It consistently captures geographical regions with elevated CO levels, with high correlation coefficients (0.58-0.95) between the model and observations and mean biases ranging from -3.92 to 5.97 DU (Table S2). The GEOS-Chem model also successfully reproduces the observed magnitude and spatial distribution of tropospheric NO<sub>2</sub> across Africa (Figure S4), particularly over equatorial Africa in January and regions like the Congo and Angola in July, with relatively small mean biases (approximately  $0.07 \times 10^{16}$  molecules cm<sup>-2</sup>).

We find that the ability of the model to reproduce TROPOMI retrievals of tropospheric ozone is noticeably worse than for CO or NO<sub>2</sub> (Figures S2 and S5). While the model reproduces the observed ozone distribution reasonably well in January and October (with correlation coefficients 0.54 and 0.80), its performance deteriorates significantly in April and July (with correlation coefficients -0.28 and -0.03). We find that limitations in the spatial coverage of TROPOMI tropospheric ozone observations, combined with the use of three-day averaging in data processing, contribute to increased uncertainty in model evaluation. We find much better agreement between our NBASE model simulation and OMI tropospheric ozone observations. Monthly correlation coefficients ranging from 0.72 to 0.92 and mean positive biases of between 2.26 and 9.17 DU. This positive model bias is likely due to the absence of averaging kernels in the monthly OMI tropospheric ozone data product. Without applying the averaging kernel, any high-resolution profile structure in the model where the satellite has low sensitivity will result in a model positive bias. The OMI/MLS tropospheric column is constructed as a residual between OMI total ozone

and MLS stratospheric ozone (Ziemke et al., 2006; Ziemke et al., 2019), and therefore does not include averaging kernel information. While OMI/Aura Level-2 datasets with averaging kernels are available, they are generally specific to the total ozone column that is less relevant to the focus of our study. In addition, satellite retrievals of tropospheric ozone are subject to relatively large uncertainties compared to CO and NO<sub>2</sub> products, partly due to retrieval sensitivity to clouds and the vertical distribution of ozone (Gaudel et al., 2018). Therefore, part of the model–satellite discrepancies in ozone may also reflect retrieval uncertainties.

Figure 4 shows Taylor diagrams for the comparison between the model runs and observations. The normalized standard deviation is represented by arcs centred at the origin, where a radius of 1 corresponds to the observed standard deviation. The arcs originating from the REF point along the horizontal axis represent the CRMSE relative to the observations. The correlation coefficient is determined by the cosine of the angle formed by a point's position vector. Points located near the REF point demonstrate the highest correlation, comparable variance to observations, and the smallest RMSE, signifying optimal performance. Analysis of the mean bias between the models and TROPOMI CO reveals that the GFAS daily emissions inventory performs best in January (0.13 DU), April (-3.41 DU), and July (1.56 DU). During October, the FREM emissions inventory has a slightly lower correlation coefficient, but its mean bias is smaller. For NO<sub>2</sub>, the FREM emission inventory performed the best with average correlation coefficients (r=0.84 for FREM hourly and FREM daily, and r=0.85 for FREM 3hourly), significantly higher than using the GFED4 (r=0.59) and GFAS inventories (r=0.62), and small mean biases relative to observed values of ~0.2×10<sup>15</sup> molecules cm<sup>-2</sup> over the four months. When compared to TROPOMI tropospheric ozone, the mean ozone biases across of FREM hourly exhibit a lower difference during high-correlation months such as January and October. Similarly, evaluation against OMI tropospheric ozone highlights the superior performance of the FREM inventory, much closer to the REF point (Figure 4), evidenced by a higher mean correlation coefficient in FREM hourly (r = 0.86) relative to GFED4 (r = 0.82) and a lower mean bias in FRME hourly compared to GFED4 during the four months (3.96) DU vs. 4.67 DU; Table S1). Additionally, we also find that the increasing the temporal resolution of the FREM inventory systematically amplifies simulation biases for CO, NO<sub>2</sub>, and tropospheric ozone columns. Nevertheless, statistical analysis confirms these differences remain insignificant (Table S1), consistent with the ozonesonde comparisons. This insensitivity partly reflects the fact that most available evaluation datasets are located outside the core fire regions of Africa, where longrange transport tends to smooth local diurnal variations. As a result, bulk evaluation metrics underestimate the impact of subdaily emission differences. Nevertheless, our sensitivity experiments demonstrate that diurnal variability exerts important mechanistic influences on ozone chemistry and transport that are not captured by fixed empirical diurnal cycles.

Overall, we find that the FREM emissions inventory demonstrates an advantage in simulating the temporal and spatial distribution of CO, NO<sub>2</sub>, and ozone. This finding, to some extent, confirms the significant value of BB emissions inventories constructed using a top-down approach with geostationary satellite observation data in atmospheric chemistry models. Because the FREM hourly and FREM daily emissions inventories performed similarly against independent satellite data, we cannot

conclude that introducing data-driven, day-specific diurnal variations improves model performance compared to the same inventory values distributed over the 24-period using an assumed diurnal cycle. Nevertheless, this result does provide an opportunity to examine how diurnal variations of BB emissions influence atmospheric composition.

# 3.3 Sensitivity of Tropospheric Ozone Chemistry over Africa to Changes in the Diurnal Variations in African BB Emissions

Here, we assess the influence of data-driven, day-specific diurnal variation of BB emissions on tropospheric ozone. This uses
the same magnitude of emissions but distributed differently over the diurnal cycle. One with the real emissions variation
derived from the observed FRP values (Nguyen et al., 2023) and the other with the same daily totals but distributed according
to a fixed diurnal cycle.

Figure 5 shows the FREM\_hourly minus FREM\_daily differences in BB emission of NO<sub>x</sub> for January, April, July, and October 2019. We find that the largest differences are observed in January and July, corresponding to the peak northern and southern hemisphere Africa fire seasons respectively. Differences are characterized by heterogeneity in space, which highlights the large uncertainty when using empirical scaling factors for prescribing BB diurnal cycles. As such, we further select two geographical regions with pronounced diurnal variability during the peak landscape fire months, as shown in Figure 5. In January, the region with significant diurnal differences is in central equatorial Africa, encompassing areas like the Central African Republic, characterized by higher daytime emissions, and South Sudan, with relatively lower daytime emissions. The regional average diurnal difference is 0.01 Gg hour<sup>-1</sup>, with a negative difference in daytime emissions and a positive difference in nighttime emissions. In July, the area with substantial diurnal variability is primarily concentrated in central southern Africa, including Angola, Zambia, and southern Congo. This geographical region exhibits the opposite trend, with positive differences in emission during the daytime and negative difference in emissions at night. The average diurnal difference is 0.03 Gg hour<sup>-1</sup>, with peak values exceeding 0.4 Gg hour<sup>-1</sup>.

FREM\_hourly minus FREM\_daily differences in the diurnal distribution of these trace gas emissions can significantly affect atmospheric composition. Figure 6 compares FREM\_hourly and FREM\_daily to assess the specific impact of diurnal variations in BB on 24-hour mean, daytime (08:00–20:00 local time), and nighttime (20:00–08:00 local time) mean values for surface and tropospheric ozone. We find that incorporating data-driven hourly resolved emissions (FREM\_hourly) leads to large differences, particularly in regions with strong diurnal BB variations (Figure 5). Differences in surface ozone levels across Africa, particularly during peak fire seasons, range from -8.57 to 21.88 ppbv. In January, central equatorial Africa exhibits notable differences in surface ozone, with a 24-hour mean difference of 0.59 ppbv and a maximum value of 6.87 ppbv. The mean differences for daytime and nighttime remain consistent at 0.69 ppbv and 0.49 ppbv, respectively. Tropospheric ozone column differences ranged from -0.37 to 0.80 DU (daytime) and -0.32 to 0.75 DU (nighttime), indicating that both daytime and nighttime emissions contribute comparably to ozone levels during this month. In July, we find more pronounced

differences in surface and tropospheric ozone over the Southern Hemisphere, particularly in Angola, where the 24-hour mean ozone differential in this region reached 0.75 DU for column and exceeded 10 ppbv for surface. The maximum diurnal difference in the tropospheric ozone column occurs at night, reaching -0.41 to 1.09 DU, compared to -0.59 to 0.66 DU during the daytime. For surface ozone, the regional average difference is higher at night (1.44 ppbv) than during the daytime (0.64 ppbv). These differences in tropospheric ozone, exclusively caused by different assumptions about the diurnal variation of BB emissions rather than the daily totals, underscore the critical importance of accurately representing the temporal structure of emissions in chemical transport models, especially for capturing local ozone dynamics during peak burning periods.

The difference in the diurnal release of emissions affects the production and loss of tropospheric ozone, which are in turn linked to the diurnal variations in the horizontal and vertical motion and mixing of the atmosphere. In particular, we find these differences in tropospheric chemistry are driven by enhanced daytime chemical production of ozone. We show in Figure 7 that January and July exhibit consistent enhancement of chemical effects in the highlighted areas, where intensified photochemistry under strong solar radiation amplifies VOC oxidation and NO<sub>x</sub> cycling efficiency. Intense landscape fires release large amounts of nitrous acid which is subsequently photolysed to produce OH radicals. In the presence of NO<sub>x</sub>, these radicals oxidize VOCs, leading to ozone formation (Xu et al., 2021). Nighttime ozone accumulation occurs under conditions where NO<sub>x</sub> emissions are lower than the empirical diurnal cycle derived from the model, owing to weakened chemical production and titration effects. The ozone titration effect, driven by NO + O<sub>3</sub>  $\rightarrow$  NO<sub>2</sub> + O<sub>2</sub>, diminishes when NO<sub>x</sub> availability declines, further exacerbating ozone accumulation (Jacob, 2000). Conversely, in regions where nighttime emissions increase and daytime emissions decrease, ozone dynamics follow an opposite pattern (Figure 6). Additionally, we find that in the focus areas for both January and July, ozone transport contributed to a net decrease in local ozone concentrations (Figure S7). This is primarily attributed to mean wind circulation—northeasterly winds in January and southeasterly winds in July—which facilitate ozone advection to downwind regions. Consequently, the diurnal variations in BB emissions not only affect local ozone levels but also influence ozone transport to downwind oceanic regions, as illustrated in Figure 6. These findings highlight the interplay between chemical processes and meteorology in shaping ozone variability under the impacts of the diurnal cycle of BB emissions over Africa. These findings are also consistent with extensive observational and modelling evidence, demonstrating that African biomass burning emissions influence downwind atmospheric chemistry through large-scale circulation patterns, following previous work (Winkler et al., 2008; Holanda et al., 2023).

# 3.4 The Impact of Improved Diurnal Variations of African BB Emissions on Global Atmospheric Chemistry

The atmosphere is a global common, so any change in our ability to describe emissions from one continent will have implications for understanding the atmospheric composition downwind of that continent. We investigate the global impact of our data-driven, day-specific diurnal variations of African BB emissions on seasonal atmospheric composition. Our focus is on surface ozone, tropospheric ozone columns, and related precursor species, including CO, formaldehyde (HCHO), NO<sub>2</sub>, and peroxyacetyl nitrate (PAN). By comparing our global model simulations that are driven by FREM, GFREM\_hourly and

GFREM\_daily, we quantify the influence of diurnal difference on these species. Figures 8–10 illustrate the impact of adopting different diurnal variations of African BB emissions by reporting seasonal, 24-hour mean differences in surface ozone, tropospheric ozone columns, tropospheric OH, and tropospheric CO, HCHO, NO<sub>2</sub>, and PAN in 2019.

# 500 Global impacts of the diurnal cycle of BB emissions over Africa on tropospheric atmospheric chemistry

BB emissions in Africa are smaller during MAM than other seasons, with the smallest diurnal difference, so discrepancies in the diurnal allocation of BB have a weaker effect on surface ozone formation, as expected. We also find that African BB emissions are more localised in the MAM and DJF than in other two seasons so that the impact on ozone is also more localised. These patterns are primarily attributed to the convergence of wind fields near the equatorial region during the MAM and DJF, which limits the spatial extent of diurnal variation effects to equatorial Africa while enabling westward transport. The surface ozone changes range from -0.75 to 0.44 ppbv during the MAM, the smallest among all seasons (Figure 8). The impact on the tropospheric ozone column during this season is also relatively minor, ranging from -0.15 to 0.07 DU (Figure 10). During DJF, the intense landscape fire activity over equatorial central Africa amplifies the impact of diurnal emission variations (Figure S8), leading to surface ozone changes ranging from -1.45 to 4.96 ppbv across the region. This enhanced variability contributes to a widespread increase in tropospheric ozone, with column enhancements reaching up to 0.53 DU.

During JJA and SON, surface ozone is more sensitive to changes in the daily patterns of African BB emissions compared to other times of the year. BB emissions during JJA show the most pronounced effect on surface ozone concentrations (Figure 8), leading to differences that range from -0.94 to 7.86 ppbv. Consistent with our high-resolution simulations, the intensified landscape fire activity during the JJA and SON in countries like Angola and Zambia plays a key role in atmospheric photochemistry (Figure S8). Elevated daytime emissions during these seasons substantially enhance ozone production, while the comparatively low nighttime emissions suppress local ozone depletion processes, including reactions with NO. This diurnal emission disparity drives a notable 24-hour surface ozone increase, reaching up to 7.9 ppbv in Angola and Zambia. The influence of diurnal differences in BB emissions extends to higher altitudes, causing a change in the tropospheric ozone column that ranges from -0.28 to 0.44 DU (Figure 10). By including data-driven diurnal variations in BB emissions results in further-reaching impacts on ozone concentrations, with detectable effects spanning regions as distant as South America, the Persian Gulf, and Australia (Figure 8). These findings emphasize the critical importance of incorporating diurnal emission variability to capture the global-scale influence of BB on atmospheric composition accurately.

Including data-driven diurnal variability of BB emissions leads to surface ozone changes of -2.90% to 13.63% relative to the empirical diurnal variability of the model in four different seasons (Figure S9). We find these differences propagate throughout the troposphere (Figure 10a). Unlike surface impacts, the impact on the free troposphere is more uniform, with their influence spreading to the Southern Hemisphere through atmospheric circulation. Importantly, Real et al. (2010) combined observational data and modelling to identify cross-hemispheric transport of biomass burning pollutants from central Africa, demonstrating

that a substantial fraction of enhanced ozone in mid-troposphere over the Atlantic primarily originate from BB sources during the summer monsoon period.

Atmospheric oxidation plays a pivotal role in determining the chemical composition of the troposphere and regulating the lifetime of key greenhouse gases and pollutants, such as methane, CO, and VOCs. Hydroxyl radicals are central to the oxidative capacity of the global troposphere. Understanding how processes such as BB influence atmospheric oxidation is crucial for predicting regional air quality, global climate dynamics, and the fate of trace gases. Figures 9 and S11 reveal how the diurnal cycle of BB emissions in Africa induces significant changes in global atmospheric oxidation. Under the influence of these diurnal differences, we find that tropospheric OH concentrations fluctuate by -12.18 × 10<sup>4</sup> to 7.94 × 10<sup>4</sup> molec cm<sup>-3</sup> (Figure 535 9), representing relative changes of -4.41% to 51.74% (Figure S11). These impacts are most pronounced in equatorial lowlatitude regions, driven by interactions between OH and the oxidation of CO, ozone, and VOCs. In South America, a region characterized by high VOC and low NO<sub>x</sub> levels, the substantial reduction in VOC concentrations, including HCHO, during BB seasons affects the OH regeneration mechanism via peroxyl radical reactions (e.g.,  $RO_2 + HO_2 \rightarrow 2OH$ ; Lu et al. (2019a)). 540 This leads to a pronounced decrease in tropospheric OH, with reductions of up to -4.41%, particularly during JJA and SON. Such findings underscore the far-reaching influence of African BB on distant regions like the Amazon rainforest, where air quality and ecosystem dynamics are closely tied to atmospheric oxidation processes. Conversely, in the high latitudes of northern and southern Africa, significant decreases in CO levels result in enhanced tropospheric OH concentrations ( $\sim 3 \times 10^4$ molec cm<sup>-3</sup>), particularly during daytime.

BB emissions substantially influence surface and near-surface ozone formation during daytime, highlighting the critical need to account for their temporal variability. While diurnal variations in BB have relatively minor global impacts on ozone concentrations, their regional effects can be profound, particularly in areas with intense fire activity where they significantly alter ozone diurnal cycles and challenge modelling accuracy. These variations also influence downwind regions along major transport pathways, altering ozone precursors and oxidative capacity far beyond Africa, potentially impacting trend interpretations (Wang et al., 2024). An additional source of uncertainty arises from the treatment of plume injection height. In this study, we used daily plume heights from GFAS, since the FREM inventory does not provide this information. The lack of sub-daily variability and uncertainties in the vertical distribution of emissions may influence the efficiency of lofting into the free troposphere, thereby modulating both the magnitude of local ozone impacts and the extent of long-range transport. Importantly, even small changes at the global scale can be relevant for the tropospheric ozone budget and radiative forcing.

Thus, the main scientific value of including geostationary-derived diurnal variability lies in capturing these regional and process-level impacts rather than producing large global mean differences.

# Mechanisms for the global impact of diurnal cycle differences in BB emissions

To investigate the mechanisms behind this impact, we further calculate the ozone budget diagnostics in the GEOS-Chem model. These diagnostics include the effects of chemistry, transport, mixing, and convection within the planetary boundary layer. Diagnostic magnitude changes suggest that chemical processes are the main drivers of ozone changes in other regions (Figure S10). This result indicates that due to the allocation of the diurnal cycle in BB emissions in Africa, coupled with the influence of wind patterns, changes in the ozone precursors from African BB are transported to regions such as the United States, the Persian Gulf, and Australia where they can affect local photochemistry. We use CO as a tracer gas to diagnose the transport processes of African BB emissions. As depicted in Figure 10b, CO concentrations move westward under the influence of easterlies near the equator, suggesting that variations in African BB significantly affect ozone precursors in North America, South America, and Australia. Our results are consistent with a previous finding that highlighted the far-reaching impacts of African BB emissions, through atmospheric transport to adjacent oceans and beyond (Sinha et al., 2004). During the dry season, approximately 60 Tg of CO is transported eastward across 0°-60°S to the Indian Ocean, while about 40 Tg CO moves westward across 0°-20°S to the Atlantic, with a substantial portion subsequently recirculating eastward through higher-latitude pathways (21°-60°S) (Sinha et al., 2004). Previous studies have also demonstrated that variations in NO<sub>x</sub> emissions are the primary driver of intercontinental transport effects on surface ozone concentrations in the northern mid-latitudes (Fiore et al., 2009). NO<sub>x</sub> can be converted into PAN through photochemical reactions involving VOCs and radicals, forming a thermally stable reservoir species that facilitates long-range transport of reactive nitrogen (Lu et al., 2019b).

For northern South America, the diurnal variation in African BB emissions results in an increase in surface ozone concentrations. We find that the increase in surface ozone is due mainly to daytime  $NO_x$  emissions over Africa being transported westward to South America by the easterlies. And because tropical South America is mainly a  $NO_x$ -limited regime with high BVOC emissions (Sun et al., 2025), this leads to a decrease in HCHO and PAN concentrations (Figure 10c, e), and a rise in surface ozone of ~0.1 ppbv and entire tropospheric ozone of ~0.05 DU. During the daytime, elevated  $NO_x$  levels are transported to the southern United States under the influence of anticyclonic airflow over the Atlantic. In the  $NO_x$ -limited southeastern United States, where VOCs (predominantly isoprene) are already saturated in most areas (Porter and Heald, 2019), this additional  $NO_x$  enhances local ozone production through accelerated photochemical reactions. In contrast, the high- $NO_x$  region of southwestern United States experiences a suppression of ozone formation due to the excess  $NO_x$  (Jin et al., 2020), which shifts the local chemical regime and reduces ozone generation efficiency (Figure S10).

For the Persian Gulf and South Asia, the westerly winds from Northern Africa, coupled with reduced BB emissions, lead to a significant diminished eastward transport of BB emissions. This phenomenon, combined with the topographic barrier of the Tibetan Plateau, exacerbates local ozone reductions. Surface ozone reductions are relatively small, around 0.2 ppbv, and we find that tropospheric ozone columns are reduced by  $\sim 0.1$  DU in some areas. This indicates that reduced ozone columns are strongly influenced by changes in the free and upper troposphere. During SON, the widespread reduction in African BB

emissions leads to decreased transport of ozone precursors as far as Australia. Compounded by the influence of weak lower-590 tropospheric cyclonic systems (Figure 8), this effect induces additional upward transport of ozone precursors, leading to localized ozone reductions of approximately -0.2 ppby in specific regions.

## **4 Summary and Conclusions**

Africa is the largest global source of BB emissions, which significantly impact atmospheric chemistry, including tropospheric ozone. BB emissions of  $NO_x$  and VOCs are key precursors for tropospheric ozone formation, with their concentrations and distribution strongly influenced by fires seasonal and diurnal dynamics. However, most conventional BB emission inventories provide emissions at daily resolution. Although some regional inventories can reach hourly resolution, high-temporal-resolution estimates remain largely unavailable over Africa. As a result, sub-daily variability is generally represented only through empirical diurnal scaling factors (Jourdain et al., 2007; Ziemke et al., 2009). We have conducted a detailed analysis to assess whether this method produced inaccuracies in modelling tropospheric ozone production and transport, using in situ tropospheric ozone measurements and the global atmospheric chemistry model GEOS-Chem fed with BB emissions from these widely used inventories, along with a newly developed 'FREM' high temporal resolution emissions inventory based on geostationary satellite observations of Fire Radiative Power, that provides emission data with resolutions up to 15 minutes (Nguyen and Wooster, 2020; Nguyen et al., 2023) and available from the EUMETSAT LSA SAF (https://lsa-saf.eumetsat.int/en/).

Our analysis reveals significant differences in the magnitude of African BB NO<sub>x</sub> emissions across inventories, with GFED4.1 reporting the highest totals (~275% and ~159% greater than GFAS and FREM, respectively), consistent with evidence suggesting GFED4 may overestimate emissions (Wang et al., 2025). Our model setup effectively captured the vertical distribution of tropospheric ozone in tropical regions, with biases ranging from -1.25 to 7.74 ppbv compared to IAGOS data and -2.38 to 5.21 ppbv relative to ozonesonde data. Compared to GFED4.1 and GFAS, FREM achieved better agreement with observations for tropospheric ozone and NO<sub>2</sub>, exhibiting lower mean biases (e.g., 3.96 DU for tropospheric ozone vs. 4.96 DU in GFED4.1) and higher correlation coefficients (e.g., r = 0.84–0.86 for tropospheric NO<sub>2</sub> and ozone columns). However, incorporating real diurnal cycle data in the FREM\_hourly inventory did not significantly enhance model performance compared to the FREM\_daily inventory, which uses an assumed diurnal cycle. Both inventories showed similar biases (-1.58 to 9.13 ppbv for FREM\_daily vs. -1.54 to 9.09 ppbv for FREM\_hourly in surface ozone) and correlations with observations, with no statistically significant differences. This limited improvement likely stems from the smoothing effect of long-range atmospheric transport, which diminishes the impact of high-resolution diurnal variations, and the fact that most available evaluation datasets are located outside the core African fire regions, where local diurnal signatures are strongest. By integrating top-down approaches and geostationary satellite data, FREM demonstrates the potential of high-resolution, data-driven inventories to improve the representation of biomass burning emissions and their impact on ozone dynamics.

Diurnal variations in African BB emissions exert a significant influence on tropospheric ozone chemistry, with pronounced regional and seasonal disparities. During peak fire seasons, the adoption of data-driven hourly emissions (FREM\_hourly) reveals substantial deviations from empirically assumed fire diurnal cycles, particularly in January (northern Africa) and July (southern Africa), leading to 24-hour surface ozone differences of –8.57 to 21.88 ppbv. Enhanced daytime chemical production elevates ozone in regions like Angola, whereas reduced nighttime titration amplifies ozone accumulation where NO<sub>x</sub> emissions are suppressed. The global impacts of these diurnal variations extend beyond Africa, altering atmospheric oxidation capacity and ozone precursor distributions across continents. Seasonal variability governs the spatial extent of these effects: localized influences dominate in MAM and DJF due to equatorial wind convergence, while JJA and SON exhibit far-reaching impacts, with surface ozone changes of –0.94 to 7.86 ppbv in fire-prone regions and tropospheric column shifts of –0.28 to 0.44 DU. Enhanced daytime NO<sub>x</sub> emissions elevate ozone in NO<sub>x</sub>-limited regions (e.g., southeastern US, northern South America) but suppress it in VOC-limited areas (e.g., southwest US). These variations influence ozone precursors like CO, NO<sub>2</sub>, HCHO, and PAN, altering ozone formation and transport across far-field regions, including South America, the Persian Gulf, and Australia.

We also find that diurnal emission variability affects global atmospheric oxidation by altering OH concentrations, with relative changes ranging from -4.41% to 51.74%, driven by interactions with VOCs, CO, and ozone. In the Amazon, reduced VOC levels weaken OH regeneration, while in high-latitude Africa, decreased CO enhances OH levels. These findings highlight the critical need to integrate realistic diurnal patterns into atmospheric models, as simplified assumptions fail to capture BB-driven ozone and oxidation dynamics. Incorporating accurate diurnal emissions will improve model precision and better predict atmospheric chemistry and climate impacts. Our work provides an improved understanding of the impact of diurnal variations in BB emissions in Africa on atmospheric composition. It helps us to better understand the mechanisms by which BB affects ozone concentrations and also offers crucial insights for formulating effective air pollution control and climate change adaptation strategies. Moreover, our findings suggest that simulations of BB emissions should consider their diurnal variations to assess their impact more accurately on atmospheric chemistry. While the improvements in bulk model performance for general wildfire simulations are modest, the main significance of our method lies in capturing hourly variability and advancing process-level understanding of atmospheric chemistry during large wildfire events.

# Data availability

TROPOMI columns are available from the Goddard Earth Sciences Data and Information Services Center (GES DISC). And the TROPOMI satellite observations of CO, NO<sub>2</sub>, and O<sub>3</sub> are archived at https://doi.org/10.5270/S5P-bj3nry0, https://doi.org/10.5270/S5P-9bnp8q8, and https://doi.org/10.5270/S5P-8aqg6um, respectively. The OMI/MLS tropospheric ozone data are available from <a href="https://acd-ext.gsfc.nasa.gov/Data\_services/cloud\_slice/new\_data.html">https://acd-ext.gsfc.nasa.gov/Data\_services/cloud\_slice/new\_data.html</a>. SEVERI FRP-PIXEL and FREM-derived BB emissions are distributed freely by the EUMETSAT LAS SAF (<a href="https://lsa-saf.eumetsat.int/en/">https://lsa-saf.eumetsat.int/en/</a>)

## 650 Author contribution

HLW, PIP, XL, SJF designed the study. HLW conducted the modelling and data analyses with contributions from WM and MJW. WM and MJW contributed to the FREM biomass burning emission dataset and its interpretation. HFW, FY, LF, and KW advised methodology and interpretation of results. HLW and PIP wrote the paper with input from all authors. All authors contributed to the discussion and improvement of the paper.

# 655 Competing interests

The contact author has declared that none of the authors has any competing interests.

# Acknowledgements

We appreciate the Atmospheric Chemistry Modeling Group at Harvard University for developing and maintaining the GEOS-Chem model. We also acknowledge the teams behind the GFEDv4.1 and GFAS biomass burning emission inventories, whose contributions have been instrumental to the success of our study, as well as the EUMETSAT LSA SAF team behind the Meteosat FRP-PIXEL product. The authors also acknowledge the European Commission, Airbus, and participating airlines (including Lufthansa, Air France, Austrian, Air Namibia, Cathay Pacific, Iberia, and China Airlines) for their ongoing support in operating and maintaining the MOZAIC (Measurements of Ozone on Airbus In-Service Aircraft) and IAGOS instrumentation since 1994. Financial support for IAGOS has been provided through the European Union's IAGOS-DS and IAGOS-ERI initiatives, along with contributions from INSU-CNRS (France), Météo-France, Université Paul Sabatier (Toulouse), and Forschungszentrum Jülich (Germany).

# Financial support

This study is supported by the National Key Research and Development Program of China (2024YFC3714200), the National Natural Science Foundation of China (42375092), and the Guangdong Basic and Applied Basic Research Foundation (2025B1515020034). WM, PIP, MJW, FY, and LF acknowledge funding from the UK National Centre for Earth Observation (NCEO) funded by the Natural Environment Research Council (NE/R016518/1, NE/Y006216/1).

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

Table 1: GEOS-Chem model simulations conducted in this study.

| Simulation   | Range  | Time period        | Biomass burning emissions     |                |  |  |
|--------------|--------|--------------------|-------------------------------|----------------|--|--|
|              | Range  | Time period        | Africa                        | Outside Africa |  |  |
| GBASE        | Global | All month          | GFED4_3hourly                 | GFED4_3hourly  |  |  |
| NBASE        | Nested | Jan, Apr, Jul, Oct | GFED4_3hourly                 | /              |  |  |
| GFAS_daily   | Nested | Jan, Apr, Jul, Oct | GFAS_daily with diurnal cycle | /              |  |  |
| FREM_hourly  | Nested | Jan, Apr, Jul, Oct | FREM_hourly                   | /              |  |  |
| FREM_3hourly | Nested | Jan, Apr, Jul, Oct | FREM_3hourly                  | /              |  |  |
| FREM_daily   | Nested | Jan, Apr, Jul, Oct | FREM_daily with diurnal cycle | /              |  |  |
| GFREM_hourly | Global | All month          | FREM_hourly                   | GFED4_3hourly  |  |  |
| GFREM_daily  | Global | All month          | FREM_daily with diurnal cycle | GFED4_3hourly  |  |  |

Table 2: Mean bias between observations and simulations (unit:ppbv) for in situ measurements (mean tropospheric ozone biases for profiles).

| Regions                       | Measurements Month GFED4_3hourly |     | GFAS_daily        | FREM_daily        | FREM_3hourly FREM_hourly |                   |                   |
|-------------------------------|----------------------------------|-----|-------------------|-------------------|--------------------------|-------------------|-------------------|
| Northern<br>Hemisphere Africa | IAGOS                            | Jan | $1.81 \pm 10.99$  | $0.96 \pm 11.21$  | $1.02 \pm 11.19$         | $0.99 \pm 11.16$  | $0.98 \pm 11.16$  |
|                               |                                  | Apr | $1.88\pm7.78$     | $1.55\pm7.70$     | $1.38\pm7.72$            | $1.37 \pm 7.72$   | $1.37\pm7.72$     |
|                               |                                  | Jul | $-1.25 \pm 6.07$  | $-2.15 \pm 5.84$  | $-1.92 \pm 5.88$         | $-1.92 \pm 5.87$  | $-1.93 \pm 5.87$  |
|                               |                                  | Oct | $7.74 \pm 4.53$   | $7.88 \pm 4.48$   | $7.73 \pm 4.54$          | $7.72 \pm 4.54$   | $7.72 \pm 4.54$   |
| Southern<br>Hemisphere Africa | IAGOS                            | Jan | $3.42 \pm 6.78$   | $3.42 \pm 6.74$   | $3.45 \pm 6.76$          | $3.41 \pm 6.74$   | $3.41 \pm 6.74$   |
|                               |                                  | Apr | /                 | /                 | /                        | /                 | /                 |
|                               |                                  | Jul | $2.57 \pm 4.41$   | $-0.67 \pm 5.01$  | $-0.12 \pm 4.88$         | $-0.29 \pm 4.89$  | $-0.37 \pm 4.91$  |
|                               |                                  | Oct | $1.77 \pm 7.66$   | $0.39 \pm 6.42$   | $-0.31 \pm 7.09$         | $-0.28 \pm 7.03$  | $-0.30 \pm 7.03$  |
| Ascension                     | Ozonesonde                       | Jan | $2.73 \pm 7.45$   | $0.50 \pm 7.42$   | $0.95 \pm 7.39$          | $0.97 \pm 7.40$   | $0.98 \pm 7.40$   |
|                               |                                  | Apr | /                 | /                 | /                        | /                 | /                 |
|                               |                                  | Jul | $5.21 \pm 8.57$   | $-0.65 \pm 8.17$  | $-0.06 \pm 7.88$         | $-0.03 \pm 7.84$  | $-0.03 \pm 7.84$  |
|                               |                                  | Oct | $4.21 \pm 11.55$  | $3.03 \pm 11.36$  | $2.85 \pm 11.35$         | $2.83 \pm 11.35$  | $2.83 \pm 11.36$  |
| Nairobi                       | Ozonesonde                       | Jan | $2.16 \pm 6.18$   | $1.48 \pm 5.84$   | $1.57 \pm 5.85$          | $1.57 \pm 5.85$   | $1.57 \pm 5.85$   |
|                               |                                  | Apr | $-2.38 \pm 9.81$  | $-2.40 \pm 9.87$  | $-2.46 \pm 9.83$         | $-2.47 \pm 9.83$  | $-2.47 \pm 9.83$  |
|                               |                                  | Jul | $2.21 \pm 5.84$   | $-0.44 \pm 5.54$  | $-0.04 \pm 5.39$         | $0.02 \pm 5.46$   | $0.02 \pm 5.46$   |
|                               |                                  | Oct | $-0.07 \pm 7.36$  | $-0.09 \pm 7.34$  | $-0.08 \pm 7.33$         | $-0.08 \pm 7.32$  | $-0.08 \pm 7.32$  |
| South Africa                  | Ground                           | Jan | $8.77 \pm 12.94$  | $8.89 \pm 12.94$  | $9.13 \pm 12.99$         | $9.09 \pm 13.02$  | $9.09 \pm 13.02$  |
|                               |                                  | Apr | $5.82 \pm 11.63$  | $5.83 \pm 11.64$  | $5.82 \pm 11.64$         | $5.82 \pm 11.64$  | $5.82 \pm 11.64$  |
|                               |                                  | Jul | $-0.93 \pm 10.66$ | $-1.70 \pm 10.58$ | $-1.58 \pm 10.66$        | $-1.54 \pm 10.67$ | $-1.54 \pm 10.66$ |
|                               |                                  | Oct | $7.96 \pm 11.81$  | $6.92 \pm 11.47$  | $6.81 \pm 11.46$         | $6.74 \pm 11.55$  | $6.73 \pm 11.54$  |

Figure 1: Comparison of the diurnal variations and monthly total NO<sub>x</sub> emissions from BB as represented by GFED4.1, GFAS, and FREM inventories in 2019. The top panel shows the diurnal cycles of NO<sub>x</sub> emissions (Gg hour<sup>-1</sup>) for January, April, July, and October, highlighting differences in diurnal patterns among the datasets. The second row displays the corresponding monthly total NO<sub>x</sub> emissions (Gg month<sup>-1</sup>), and third-to-fifth row is their spatial distribution in 2019 along with the total emission for that year.

Figure 2: Evaluation of simulated ozone against multi-platform observations over Africa in 2019. (a) Spatial distribution of measurement platforms including IAGOS aircraft flight tracks (color-coded by ozone concentrations), ozonesonde launch sites (triangles), and surface monitoring stations (square). Vertical profile comparisons between IAGOS aircraft observations and model simulations for (b) Northern Hemisphere and (c) Southern Hemisphere Africa. (d, e) Ozonesonde profile comparisons at (d) Ascension and (e) Nairobi sites, with horizontal bars indicating standard deviations in the observations (±1σ). (f-i) Temporal evolution of surface ozone concentrations across 84 monitoring stations in South Africa for January (f), April (g), July (h), and October (i), showing observed (black lines for IAGOS observation and dots for surface observation) and simulated (colored lines) values using different BB emission inventories.

Figure 3: Mean biases in GEOS-Chem simulated ozone concentrations compared with observations from (a) ground-based measurements in South Africa, (b) aircraft observations over Northern Hemisphere Africa (NHAF), (c) aircraft observations over Southern Hemisphere Africa (SHAF), and ozonesonde measurements at (d) Ascension and (e) Nairobi. The error bars represent ±1/2 standard deviation of the mean biases. The data are categorized by month (Jan, Apr, Jul, Oct) and differentiated by BB emission inventory (GFAS daily, GFED4 3hourly, FREM daily, FREM 3hourly, FREM hourly).

Figure 4: Comparison of Taylor statistics of CO, NO<sub>2</sub>, and ozone, simulated using the different BB emissions. The REF point on the horizontal axis denotes the reference observations, representing perfect agreement with normalized standard deviation = 1, correlation coefficient = 1, and centred RMSE = 0.

Figure 5: Spatial distribution and frequency of diurnal differences between FREM hourly BB emissions and FREM daily emissions (FREM\_hourly - FREM\_daily) for January, April, July, and October 2019. The top panels illustrate the spatial distribution of NO<sub>x</sub> emission differences (Gg hour<sup>-1</sup>) during daytime and nighttime. The bottom histograms show the frequency distributions of these differences in highlighted areas of high wildfire intensity in January and July.

Figure 6: Impact of diurnal variations in BB emissions on GEOS-Chem model estimates of surface and tropospheric ozone concentrations over Africa. The panels illustrate the differences between the FREM\_hourly and FREM\_daily schemes (FREM\_hourly - FREM\_daily) for January and July, with results presented for 24-hour mean, daytime mean, and nighttime mean. Surface ozone differences are highlighted in the inset maps, while vectors represent the 850 hPa wind fields.

Figure 7: Impact of diurnal variations in BB emissions on ozone budget diagnostics (kg h<sup>-1</sup>) in Africa for January and July 2019, as estimated by the GEOS-Chem model. The diagnostics include contributions from chemistry, transport, convection, and mixing, with results shown separately for daytime and nighttime in each month. The first column presents the highlighted regional mean values of daytime and nighttime (gray shading) ozone budget terms for January and July, while the subsequent columns display their spatial distributions.

Figure 8: Impact of diurnal variations in BB emissions in Africa on global surface ozone concentrations and seasonal mean wind fields at 850 hPa, as estimated by the GEOS-Chem model for different seasons in 2019. The difference between the sensitivity experiments GFREM\_hourly and GFREM\_daily represents the impact of diurnal variation (GFREM\_hourly - GFREM\_daily).

Figure 9: Impact of diurnal variations in BB emissions in Africa on global tropospheric OH estimated by GEOS-Chem for different seasons in 2019. The difference between the sensitivity experiments GFREM\_hourly and GFREM\_daily represents the impact of diurnal variation (GFREM\_hourly - GFREM\_daily).

Figure 10: Impact of diurnal variations in BB emissions in Africa on global tropospheric ozone (top row; a), CO (second row; b), HCHO (third row; c), NO<sub>2</sub> (forth row; d), and PAN (fifth row; e) estimated by GEOS-Chem for different seasons in 2019. The difference between the sensitivity experiments GFREM\_hourly and GFREM\_daily represents the impact of diurnal variation (GFREM\_hourly - GFREM\_daily).