# Peer review of "Using Geostationary-Satellite-Derived Sub-Daily Fire Radiative Power Variability versus Prescribed Diurnal Cycles to Assess the Impact of African Fires on Tropospheric Ozone"

_EGUsphere, 2025_

## Author Comment (AC1)

**Dear Editor Dr. James Lee,**

**Thank you very much for handling our manuscript. Please find below our itemized responses to the reviewers' comments and a marked-up manuscript. We have addressed all the comments raised by the reviewer and incorporated them in the revised manuscript.**

**Thank you for your consideration.**

**Sincerely,**
**Haolin Wang et al.**
* * *
**Reviewer #1**

**Comment [1-1]:** Wang et al. aim to enhance the temporal resolution of current biomass burning emission estimates, which are typically provided on a daily to monthly basis. When integrated into chemical transport models, these emission estimates currently rely on fixed diurnal cycles informed by climatological empirical data. The authors introduce a new method that utilizes day-specific diurnal variations derived from geostationary Fire Radiative Power (FRP). They then examine how these improvements in emissions impact simulated ozone levels. Overall, the subject matter aligns with the scope of ACP. The presentation is well-structured and clear, though the resulting improvements are subtle.

**Response [1-1]:** We thank the reviewer for the positive comments on our study. Below, we provide a point-by-point response to the reviewer's comments and summarize the changes that have been made in the revised manuscript.

**Comment [1-2]:** Abstract, L21-23: The authors' new method of incorporating day-specific diurnal variations from geostationary observations, compared to fixed diurnal cycles based on climatological data, appears to yield only subtle improvements. The surface ozone biases are reduced slightly from '−1.54 to +9.09 ppbv' to '−1.58 to +9.13 ppbv'. This is particularly noteworthy given that significant changes to ozone levels can be achieved by addressing other uncertainties in wildfire emissions (e.g., injection height, emission budgets, etc.). The diurnal cycles of NOx seem to be reasonably captured by the diurnal scaling factors applied in both the GFED4 and GFAS inventories (Fig. 1), with no significant differences in bias between the new method and the conventional GFED and GFAS approaches (Fig. 3). How does this new method improve model performance or enhance our general understanding of current knowledge?

**Response [1-2]:** We thank the reviewer for this constructive comment. We agree that the overall statistical improvement in surface ozone biases is small. However, the principal contribution of our approach is not simply to reduce the bias, but to provide a more physically realistic representation of the diurnal variability of biomass burning emissions. Our simulations show that the timing and amplitude of diurnal NOx emissions differ substantially between the geostationary-derived inventory and conventional climatological scaling factors. While these differences do not strongly alter domain-wide mean biases due to the smoothing effect of long-range transport, and the fact that most available evaluation datasets are located outside the core African fire regions, they lead to pronounced local and regional impacts on ozone (up to −8.57 to +21.88 ppbv at the surface and −0.41 to +1.09 DU in tropospheric columns) and on OH concentrations (−4.4% to +51.7%) during the fire season. These effects propagate to remote regions through atmospheric circulation, influencing chemical regimes as far

afield as South America and the Persian Gulf. Thus, our method enhances our understanding of how sub-daily fire activity modulates atmospheric chemistry and demonstrates the importance of integrating realistic diurnal variability in future global models. We have revised the text to highlight these mechanistic insights in addition to the statistical evaluation as follows.

L22–26: "Simulations with real hourly-resolved emissions produce comparable overall surface ozone biases (−1.54 to +9.09 ppbv vs. −1.58 to +9.13 ppbv) and marginally higher correlations with TROPOMI nitrogen dioxide (r = 0.80–0.89) and OMI ozone (r = 0.80–0.94). Although the statistical improvements are limited, the geostationary-driven approach reveals pronounced regional ozone differences and mechanistic insights into the role of diurnal fire variability."

L477–481: "This insensitivity partly reflects the fact that most available evaluation datasets are located outside the core fire regions of Africa, where long-range transport tends to smooth local diurnal variations. As a result, bulk evaluation metrics underestimate the impact of sub-daily emission differences. Nevertheless, our sensitivity experiments demonstrate that diurnal variability exerts important mechanistic influences on ozone chemistry and transport that are not captured by fixed empirical diurnal cycles."

L684–686: "This limited improvement likely stems from the smoothing effect of long-range atmospheric transport, which diminishes the impact of high-resolution diurnal variations, and the fact that most available evaluation datasets are located outside the core African fire regions, where local diurnal signatures are strongest."

**Comment [1-3]:** L229: Unclear how CO relates to other emitted species. By a fixed CO-to-species ratio for all three emission inventories? This is an important point to clarify before interpreting the results in Section 3.1 where intra-inventory NOx emissions are compared.

**Response [1-3]:** We thank the reviewer for this valuable comment and agree that the distinction between the inventories requires clearer explanation. In our study, the treatment of CO and other species differs among the three biomass burning inventories as follows:

- FREM: The FREM inventory provides top-down estimates of CO emissions based on geostationary SEVIRI FRP and Sentinel-5P CO observations (Nguyen et al., 2023). Other species (e.g., NOx, VOCs, aerosols) are derived from CO using biome-specific CO-to-species emission factor ratios reported in Andreae and Merlet (2001), Akagi et al. (2011), and Andreae (2019), consistent with Nguyen et al. (2023). This approach ensures that the conversion accounts for differences among vegetation types (e.g., savanna, tropical forest).
- GFED4.1s: GFED is a bottom-up inventory that combines MODIS burned area, fuel load estimates, combustion completeness, and biome-specific emission factors defined per unit of dry matter burned (Randerson et al., 2017; van der Werf et al., 2017). Therefore, it directly provides emissions for multiple species, which we use without any rescaling from CO.
- GFAS: GFAS assimilates MODIS FRP observations and converts them to dry matter combustion using land-cover-specific FRP-to-dry-matter-consumed factors that relate MODIS FRP to dry matter combustion rates as in GFED (Kaiser et al., 2012). It then estimates emissions of trace gases and aerosols using land-class-dependent emission factors.

Accordingly, we have revised Section 2.5 (Lines 278–285) to explicitly clarify this distinction as follows: "In GFED4.1s, emissions of trace gases and aerosols are derived using a bottom-up approach that combines MODIS

burned area, fuel load, combustion completeness, and biome-specific emission factors defined per unit of dry matter burned (Randerson et al., 2017; van der Werf et al., 2017). GFAS follows a similar principle, converting MODIS FRP into dry matter combustion using land-cover-specific FRP-to-dry-matter-consumed factors, and then deriving emissions using biome-dependent emission factors (Kaiser et al., 2012). By contrast, the FREM inventory provides only CO emissions derived from geostationary FRP and Sentinel-5P CO observations (Nguyen et al., 2023); other species (e.g., NOx, VOCs, aerosols) are obtained by scaling CO using biome-specific CO-to-species emission factor ratios (Andreae and Merlet, 2001; Akagi et al., 2011; Andreae, 2019)."

**References**

Andreae, M. O., and Merlet, P.: Emission of trace gases and aerosols from biomass burning, Global Biogeochem. Cycles, 15, 955–966, 2001.

Akagi, S. K., et al.: Emission factors for open and domestic biomass burning for use in atmospheric models, Atmos. Chem. Phys., 11, 4039–4072, 2011.

Andreae, M. O.: Emission of trace gases and aerosols from biomass burning – an updated assessment, Atmos. Chem. Phys., 19, 8523–8546, 2019.

Kaiser, J. W., et al.: Biomass burning emissions estimated with a global fire assimilation system based on observed fire radiative power, Biogeosciences, 9, 527–554, 2012.

Randerson, J. T., et al.: Global Fire Emissions Database, Version 4.1 (GFEDv4), ORNL DAAC, 2017.

van der Werf, G. R., et al.: Global fire emissions estimates during 1997–2016, Earth Syst. Sci. Data, 9, 697–720, 2017.

Nguyen, H. M., He, J., and Wooster, M. J.: Biomass burning CO, PM and fuel consumption per unit burned area estimates derived across Africa using geostationary SEVIRI fire radiative power and Sentinel-5P CO data, Atmos. Chem. Phys., 23, 2089–2118, 2023.

**Comment [1-4]:** L300 & L567: The authors may want to clarify the intra-inventory differences in the species budgets for CO, PM, and other species, in addition to the NOx presented. This would help readers better understand the full context of the intra-inventory differences and make sense of the comparisons being made.

**Response [1-4]:** We thank the reviewer for this helpful suggestion. In the main text we focus on NOx because of its direct role in ozone formation, while CO is the principal species used for intercomparison across multiple biomass burning emission inventories and serves as a key tracer of combustion and long-range transport. To provide a broader context, we have therefore included an analysis of the intra-inventory differences for CO in the Supplement (Figure S1). The CO results exhibit similar patterns to those of NOx, supporting the robustness of our conclusions.

We have clarified in the text: "A similar comparison for CO emissions is presented in Figure S1. CO is the principal species used for intercomparison across multiple biomass burning emission inventories (Pan et al., 2020; Jin et al., 2024) and serves as a key tracer of combustion and long-range transport (Duncan et al., 2003; Edwards et al., 2006). All three inventories capture the pronounced north–south seasonal contrast of African fires, with peak activity in January and July and relatively weak emissions in April and October. The total emitted CO differs noticeably among inventories, with GFED4 generally reporting higher regional totals than the others, while the relative differences vary somewhat between months and regions. The diurnal variation of CO emissions is broadly consistent with that of NOx, showing a clear daytime maximum around local noon and minima at

night, but the amplitude and peak timing vary among inventories, reflecting differences in the representation of fire activity and emission factors."

[Figure]

Figure S1. Same with Figure 1, but for CO emission.

**Comment [1-5]:** Section 3.4: Most global effects are minor, around ± 0.1-1 ppbv in Fig.8.

**Response [1-5]:** We thank the reviewer for this comment. We agree that most global mean surface ozone changes shown in Fig. 8 are relatively small (typically ±0.1–1 ppbv). This reflects the fact that the diurnal variability of African fire emissions is a sub-daily redistribution of the same daily emission totals, so the largest impacts are expected locally and seasonally rather than in global mean fields. Nevertheless, our simulations show that in regions with intense fire activity (e.g., Angola and Zambia) and along major outflow pathways, the differences are much larger (up to 7–8 ppbv at the surface and ±0.4 DU in tropospheric columns). Furthermore, even small global-scale changes of ~0.5 ppbv can be significant when integrated over the free troposphere, where

they affect the oxidative capacity and radiative forcing of ozone. We have revised the text in Section 3.4 to clarify that while global mean effects are generally modest, the regional and process-level impacts are substantial and provide the main scientific value of including realistic diurnal variability.

We have clarified in the text: "These variations also influence downwind regions along major transport pathways, altering ozone precursors and oxidative capacity far beyond Africa, potentially impacting trend interpretations (Wang et al., 2024). Importantly, even small changes at the global scale can be relevant for the tropospheric ozone budget and radiative forcing. Thus, the main scientific value of including geostationary-derived diurnal variability lies in capturing these regional and process-level impacts rather than producing large global mean differences."

**Comment [1-6]:** Conclusion: The new method may be more significant for understanding the hourly variations in atmospheric chemistry that occur throughout the diurnal progression of large wildfires. However, for general wildfire simulations, the improvements in the model are less noticeable.

**Response [1-6]:** We thank the reviewer for this helpful suggestion. We agree that the overall improvements in bulk model performance for general wildfire simulations are modest. However, the primary value of our method lies in providing a more physically realistic description of the diurnal variability of fire emissions, which is particularly important for understanding the hourly progression of atmospheric chemistry during large wildfire events.

We have clarified in the text: "Moreover, our findings suggest that simulations of BB emissions should consider their diurnal variations to assess their impact more accurately on atmospheric chemistry. While the improvements in bulk model performance for general wildfire simulations are modest, the main significance of our method lies in capturing hourly variability and advancing process-level understanding of atmospheric chemistry during large wildfire events."

**Comment [1-7]:** Title: Define the acronym FRP.

**Response [1-7]:** We thank the reviewer for this comment. We have revised the title.

**Comment [1-8]:** Fig.4: Define the acronym REF in the figure caption.

**Response [1-8]:** We thank the reviewer for pointing this out. We have revised the caption of Fig. 4 to define the acronym.

We have clarified in the text: "The REF point on the horizontal axis denotes the reference observations, representing perfect agreement with normalized standard deviation = 1, correlation coefficient = 1, and centred RMSE = 0."

---

## Author Comment (AC2)

**Dear Editor Dr. James Lee,**

**Thank you very much for handling our manuscript. Please find below our itemized responses to the reviewers' comments and a marked-up manuscript. We have addressed all the comments raised by the reviewer and incorporated them in the revised manuscript.**

**Thank you for your consideration.**

**Sincerely,**
**Haolin Wang et al.**
* * *
**Reviewer #2**

**Comment [2-1]:** This study tested how diurnal variation of fire emissions in Africa impact tropospheric ozone using GEOS-Chem. The issues below need to be addressed before it can be published.

**Response [2-1]:** We thank the reviewer for the comments on our study. Below, we provide a point-by-point response to the reviewer's comments and summarize the changes that have been made in the revised manuscript.

**Comment [2-2]:** Line 16: "GFED and GFAS" add full name.

**Response [2-2]:** We thank the reviewer for pointing this out. We have revised the text.

**Comment [2-3]:** Line 22: "reduce surface ozone biases (−1.54 to +9.09 ppbv vs. −1.58 to +9.13 ppbv)" This change is very small. Is it statistically significant? Otherwise please remove this statement.

**Response [2-3]:** We thank the reviewer for bringing this up.

We now state in the text: "Simulations with real hourly-resolved emissions produce comparable surface ozone biases (−1.54 to +9.09 ppbv vs. −1.58 to +9.13 ppbv) and marginally higher correlations with TROPOMI nitrogen dioxide (r = 0.80–0.89) and OMI ozone (r = 0.80–0.94). Although the statistical improvements are limited, the geostationary-driven approach reveals pronounced regional ozone differences and mechanistic insights into the role of diurnal fire variability"

**Comment [2-4]:** Title: "Using Geostationary-Derived Sub-Daily FRP Variability vs. Prescribed Diurnal Cycles" reads awkward. Using FRP Variability for what? Also Geostationary-Derived should be Geostationary-Satellite-Derived. Please revise.

**Response [2-4]:** Thank you for pointing it out.

We have revised the title as follows: "Using Geostationary-Satellite-Derived Sub-Daily Fire Radiative Power Variability versus Prescribed Diurnal Cycles to Assess the Impact of African Fires on Tropospheric Ozone"

**Comment [2-5]:** Line 70-72: repeated statements.

**Response [2-5]:** We thank the reviewer for pointing this out. We have removed the repeated statements in the revised manuscript.

**Comment [2-6]:** Line 150: remove a "both".

**Response [2-6]:** Thank you for pointing it out. We have removed the "both" in the revised manuscript.

**Comment [2-7]:** Line 296: "We also show emissions in the lower part of the fire seasons" not sure what this means.

**Response [2-7]:** We thank the reviewer for noting the unclear wording. The intended meaning was to indicate months with relatively low fire activity during the fire season.

We have revised the sentence in Line 351-352 to: "For comparison, we also present emissions for April and October, which represent months of relatively low fire activity within the fire season."

**Comment [2-8]:** Line 82: "Most satellite instruments, operating in sun-synchronous orbits, sample at fixed local times, limiting the temporal resolution of inventories like GFED and GFAS to weekly or monthly scales, with sub-weekly estimates relying on empirical scaling factors" This statement is wrong. GFED has monthly temporal resolution because it is based on monthly MODIS burned area product, not because of the satellite overpass time. Also is GFAS really weekly product? GFAS is FRP based product and it's hard to imagine it's a weekly/monthly dataset. Even so, there are widely used global fire emission inventories that are daily such as QFED and FINN. The authors ignored that.

**Response [2-8]:** We thank the reviewer for this important clarification. We agree that our original wording was inaccurate. Specifically, GFED has monthly temporal resolution because it is based on the MODIS burned area product provided at monthly intervals, not because of satellite overpass times. GFAS, in turn, is an FRP-based product that provides daily emissions, rather than a weekly or monthly dataset as we originally implied. We have corrected these statements in Section 2.2. In addition, we have now acknowledged other widely used global fire emission inventories such as QFED (Darmenov and da Silva, 2013) and FINN (Wiedinmyer et al., 2011), which also provide daily emissions.

We have rephrased the text as: "Most widely used global fire emission inventories, such as Global Fire Emission Database (GFED), Global Fire Assimilation System (GFAS), Quick Fire Emissions Dataset (QFED), and Fire Inventory from NCAR (FINN), provide emissions at monthly or daily resolution. GFED is based on the MODIS burned area product available at monthly intervals, while GFAS, QFED, and FINN use active fire detections or FRP to provide daily emissions that can be further disaggregated to sub-daily scales using empirical or observation-based factors (WRAP, 2005; Akagi et al., 2011; Wooster et al., 2021)"

**References**

Darmenov, A., and da Silva, A.: The Quick Fire Emissions Dataset (QFED) – Documentation of versions 2.1, 2.2 and 2.4, NASA Technical Report Series on Global Modeling and Data Assimilation, NASA/TM-2013-104606, Vol. 32, 183 pp., 2013.

Wiedinmyer, C., Akagi, S. K., Yokelson, R. J., Emmons, L. K., Al-Saadi, J. A., Orlando, J. J., and Soja, A. J.: The Fire INventory from NCAR (FINN): a high resolution global model to estimate the emissions from open burning, Geosci. Model Dev., 4, 625–641, https://doi.org/10.5194/gmd-4-625-2011, 2011.

**Comment [2-9]:** The motivation to study diurnal variation of fire emissions on atmospheric chemistry is valid. However the study overlooked previous efforts on this topic. I suggest revise the introduction to reflect previous efforts in the area. The manuscript also misses important references. Below are a few examples.

Andela, N., Kaiser, J. W., van der Werf, G. R., & Wooster, M. J. (2015). New fire diurnal cycle characterizations to improve fire radiative energy assessments made from MODIS observations. Atmospheric Chemistry and Physics, 15, 8831–8846. https://doi.org/10.5194/acp-15-8831-2015

Li, F., Zhang, X., Roy, D. P., & Kondragunta, S. (2019). Estimation of biomass-burning emissions by fusing the fire radiative power retrievals from polar-orbiting and geostationary satellites across the conterminous United States. Atmospheric Environment, 211, 274–287. https://doi.org/10.1016/j.atmosenv.2019.05.017

Tang, W., Emmons, L. K., Buchholz, R. R., Wiedinmyer, C., Schwantes, R. H., He, C., Kumar, R., Pfister, G. G., Worden, H. M., Hornbrook, R. S., Apel, E. C., Tilmes, S., Gaubert, B., Martinez-Alonso, S.-E., Lacey, F., Holmes, C. D., Diskin, G. S., Bourgeois, I., Peischl, J., Ryerson, T. B., Hair, J. W., Weinheimer, A. J., Montzka, D. D., Tyndall, G. S., Campos, T. L., Effects of fire diurnal variation and plume rise on US air quality during FIREX‐AQ and WE‐CAN based on the Multi‐Scale Infrastructure for Chemistry and Aerosols (MUSICAv0). Journal of Geophysical Research: Atmospheres, p.e2022JD036650, 2022.

Freeborn, P. H., Wooster, M. J., Roberts, G., Malamud, B. D., & Xu, W. (2009). Development of a virtual active fire product for Africa through a synthesis of geostationary and polar orbiting satellite data. Remote Sensing of Environment, 113, 1700–1711. https://doi.org/10.1016/j.rse.2009.03.013

Mu, M., Randerson, J. T., van der Werf, G. R., Giglio, L., Kasibhatla, P., Morton, D., et al. (2011). Daily and 3-hourly variability in global fire emissions and consequences for atmospheric model of predictions of carbon monoxide. Journal of Geophysical Research, 116, D24303. https://doi.org/10.1029/2011JD016245.

**Response [2-9]:** We thank the reviewer for this valuable comment. We agree that the Introduction did not sufficiently acknowledge previous work on the diurnal variability of fire emissions. We have revised the Introduction accordingly and added the suggested references.

L100-102: "More recently, Tang et al. (2022) emphasized that fire diurnal cycles exert strong influences on regional air quality during major field campaigns in the United States, underscoring the need for improved representation of sub-daily variability in chemical transport models (Mu et al., 2011; Li et al., 2019)."

L117-120: "Previous efforts have sought to characterize fire diurnal cycles directly from satellite observations, for example by combining geostationary and polar-orbiting data for Africa (Freeborn et al., 2009) and by developing new MODIS-based diurnal parameterizations to improve FRP-derived fire energy estimates (Andela et al., 2015)."

**Comment [2-10]:** Line 115: remove "high spatial resolution".
**Response [2-10]:** Corrected.

**Comment [2-11]:** Line 454: Perhaps "meteorology" or "transport" is more appropriate here than "atmospheric circulation" as "atmospheric circulation" often refers to large scale motion.
**Response [2-11]:** We thank the reviewer for this suggestion. We have revised the wording accordingly.

**Comment [2-12]:** Figure 2: do panels b-e have the same legend as panels f-i?

**Response [2-12]:** We thank the reviewer for pointing this out. Panels b–e and f–i share the same legend. We have clarified this in the caption of Figure 2 to avoid confusion.

**Comment [2-13]:** Satellite retrievals of O3 are subject to relatively large uncertainties compared to satellite CO and NO2 products. That need to be considered when you explain model-satellite discrepancies.

**Response [2-13]:** We thank the reviewer for this valuable comment. We agree that satellite retrievals of $O_3$ are subject to larger uncertainties compared to CO and $NO_2$ products. We have revised the discussion to explicitly acknowledge this limitation. Some of the observed model–satellite differences for ozone may partly reflect retrieval uncertainties.

We have clarified in the text: "In addition, satellite retrievals of tropospheric ozone are subject to relatively large uncertainties compared to CO and $NO_2$ products, partly due to retrieval sensitivity to clouds and the vertical distribution of ozone (Gaudel et al., 2018). Therefore, part of the model–satellite discrepancies in ozone may also reflect retrieval uncertainties."

**References**

Gaudel, A., Cooper, O. R., et al.: Tropospheric Ozone Assessment Report: Present-day distribution and trends of tropospheric ozone relevant to climate and global atmospheric chemistry model evaluation, Elem. Sci. Anth., 6, 39, https://doi.org/10.1525/elementa.291, 2018.

**Comment [2-14]:** Line 375: It's surprising there is no AK for OMI tropospheric O3 column. Is this true?

**Response [2-14]:** We thank the reviewer for this question. In this study we used the OMI/MLS tropospheric ozone column product (Ziemke et al., 2006; Ziemke et al., 2019), which is constructed by subtracting the co-located MLS stratospheric ozone column from the OMI total ozone column. This product provides monthly mean fields but does not include averaging kernel information, since it is not a direct profile retrieval but a residual-based column estimate. We have clarified this point in the revised manuscript.

We have clarified in the text: "The OMI/MLS tropospheric column is constructed as a residual between OMI total ozone and MLS stratospheric ozone (Ziemke et al., 2006; Ziemke et al., 2019), and therefore does not include averaging kernel information. While OMI/Aura Level-2 datasets with averaging kernels are available, they are generally specific to the total ozone column that is less relevant to the focus of our study."

**References**

Ziemke, J. R., Chandra, S., Duncan, B. N., Froidevaux, L., Bhartia, P. K., Levelt, P. F., and Waters, J. W.: Tropospheric ozone determined from Aura OMI and MLS: Evaluation of measurements and comparison with the Global Modeling Initiative's Chemical Transport Model, J. Geophys. Res., 111, D19303, https://doi.org/10.1029/2006JD007089, 2006.

Ziemke, J. R., Chandra, S., Labow, G. J., Bhartia, P. K., Froidevaux, L., and Witte, J. C.: A global ozone climatology from OMI and MLS measurements for the decade 2004–2014, Atmos. Chem. Phys., 19, 3257–3269, https://doi.org/10.5194/acp-19-3257-2019, 2019.

**Comment [2-15]:** Line 558: Again it is not fair to say "conventional BB emission inventories like GFED and GFAS are typically available at daily-to-monthly temporal resolutions". Most global fire emission inventories are daily. Some regional ones are hourly, just not over Africa.

**Response [2-15]:** We thank the reviewer for this helpful comment. GFED provides emissions at monthly resolution based on MODIS burned area, whereas GFAS, QFED, and FINN provide daily emissions, and some regional products even offer hourly resolution. In particular, we now state that while most global fire emission inventories provide daily emissions, and some regional products achieve hourly resolution, high-temporal-resolution inventories remain lacking for Africa, which motivates to make use of geostationary observations over Africa to examine how the diurnal variability of fire emissions influences ozone.

We have revised the text to clarify this point as follows: "However, most conventional BB emission inventories provide emissions at daily resolution. Although some regional inventories can reach hourly resolution, high-temporal-resolution estimates remain largely unavailable over Africa. As a result, sub-daily variability is generally represented only through empirical diurnal scaling factors."

**Comment [2-16]:** The authors need to discuss how uncertainties in plume injection height influence the results, especially for the "global impacts of the diurnal cycle of BB emissions over Africa" part.

**Response [2-16]:** We thank the reviewer for this valuable comment. The FREM high-temporal-resolution emissions inventory developed for Africa does not provide plume injection height information. Therefore, in this study we used the daily plume injection height fields from GFAS to drive the model. We acknowledge that uncertainties in plume rise remain an important limitation, as higher or lower injection heights can significantly alter the partitioning of emissions between the boundary layer and free troposphere, thereby affecting both regional ozone formation and the long-range transport of fire plumes. We have added a note in Section 3.4 to clarify this limitation when interpreting the global impacts of the diurnal cycle of BB emissions.

We have clarified in the text: "An additional source of uncertainty arises from the treatment of plume injection height. In this study, we used daily plume heights from GFAS, since the FREM inventory does not provide this information. The lack of sub-daily variability and uncertainties in the vertical distribution of emissions may influence the efficiency of lofting into the free troposphere, thereby modulating both the magnitude of local ozone impacts and the extent of long-range transport."